# ACCELERATED SAMPLING WITH STACKED RESTRICTED BOLTZMANN MACHINES

**Jorge Fernandez-de-Cossio-Diaz**[*]
jfdecd@icloud.com

**Clément Roussel**[*]
clement.roussel.2014@polytechnique.org

**Simona Cocco**
simona.cocco@phys.ens.fr

**Rémi Monasson**
remi.monasson@phys.ens.fr

Laboratory of Physics of the Ecole Normale Supérieure.
CNRS UMR 8023 & PSL Research, Sorbonne Université.
24 rue Lhomond, 75005 Paris, France. $(*)$ equal contribution

## ABSTRACT

Sampling complex distributions is an important but difficult objective in various fields, including physics, chemistry, and statistics. An improvement of standard Monte Carlo (MC) methods, intensively used in particular in the context of disordered systems, is Parallel Tempering, also called replica exchange MC, in which a sequence of MC Markov chains at decreasing temperatures are run in parallel and can swap their configurations. In this work we apply the ideas of parallel tempering in the context of restricted Boltzmann machines (RBM), a paradigm of unsupervised architectures, capable to learn complex, multimodal distributions. Inspired by Deep Tempering, an approach introduced for deep belief networks, we show how to learn on top of the first RBM a stack of nested RBMs, using the representations of a RBM as 'data' for the next one along the stack. In our Stacked Tempering approach the hidden configurations of a machine can be exchanged with the visible configurations of the next one in the stack. Replica exchanges between the different RBMs is facilitated by the increasingly clustered representations learnt by deeper RBMs, allowing for fast transitions between the different modes of the data distribution. Analytical calculations of mixing times in a simplified theoretical setting shed light on why Stacked Tempering works, and how hyperparameters, such as the aspect ratios of the RBMs and weight regularization should be chosen. We illustrate the efficiency of the Stacked Tempering method with respect to standard and replica exchange MC on several datasets: MNIST, in-silico Lattice Proteins, and the 2D-Ising model.

## 1 INTRODUCTION

**General introduction.** Sampling complex energy landscapes is an important goal in many scientific fields, including, but not limited to computational physics and statistical inference. Following the introduction of Monte Carlo (MC) methods by Metropolis in the late 40's, several approaches have been considered to speed up sampling. Among them cluster-based algorithms employ non-local moves in the configuration space that are able to update a large, possibly extensive number of microscopic variables at once (Swendsen & Wang, 1987; Wolff, 1989), but require detailed knowledge of the energy landscape for the construction of effective clusters. Another improvement of standard MC methods is parallel tempering(Swendsen & Wang, 1986), which consists in simulating several copies of the system at higher temperatures, where energetic barriers are easier to cross. Exchanges of configurations between copies of the system at different temperatures are proposed, resulting in the original system benefiting from the capability of high-temperature ones to quickly explore the configuration space, instead of getting indefinitely stuck in energetic valleys.

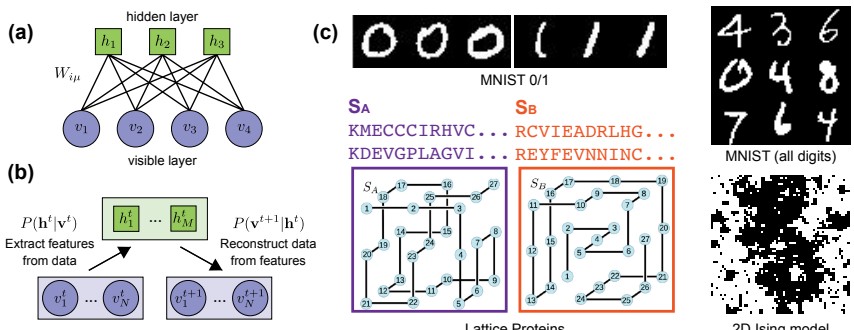

Figure 1: **(a)** RBM architecture, **(b)** Alternating Gibbs Sampling procedure. **(c)** Datasets used in the paper: MNIST digits (0/1 and all), Lattice Proteins (showing example sequences with high probability $P_{\text{nat}}$ of folding into structures $S_A$ and $S_B$, respectively), and 2-dimensional Ising model.

**Contributions of this work.** We hereafter report on an extension of the ideas of parallel tempering to data-driven modeling, which we called stacked tempering (ST). We use RBMs, a paradigm of unsupervised architectures known to be universal approximators of data distributions Roux & Bengio (2008). The distributions learnt by RBM may be complex and multimodal, and hence difficult to sample. We propose to learn stacks of nested RBMs, using the representations of a RBM as 'data' for the next one along the stack. Informally speaking, these RBMs learn more and more simplified versions of the data distributions, which become increasingly easier to sample with standard MC dynamics. We then couple the RBMs by allowing them to exchange configurations. These exchanges are made possible by the nested structure of the stack, *i.e.* the compatibility between the sizes of the layers of contiguous RBMs. We show numerically on three data sets, MNIST (LeCun, 1998), *in silico* proteins (Shakhnovich & Gutin, 1990), and the 2D Ising model Onsager (1944) that the resulting ST procedure is much faster than parallel tempering MC for sampling the learnt distributions. In addition, we present a mathematically tractable setup, in which the exponential (in the dimension of the data space) speed-up in sampling offered by ST with respect to standard Gibbs sampling is exactly calculated. We show how this speed up is intimately connected to how representations associated to data points evolve along the stack of RBM, depending on structural hyperparameters.

**Relation to previous work.** Neural network based algorithms are now widely applied to sample complex landscapes Wu et al. (2019); Noé et al. (2019); Gabrié et al. (2021). Our work is inspired by the deep tempering procedure of Desjardins et al. (2014), introduced to accurately train deep belief networks Hinton & Salakhutdinov (2006); Hinton et al. (2006); Salakhutdinov & Murray (2008). In their original work, Desjardins et al. (2014) RBM layers with similar numbers of hidden units are all trained simultaneously with the aim of improving mixing of the MC chains used in the learning algorithm Tieleman (2008). Contrary to deep tempering, stacked tempering trains RBMs one at a time, and is easier to implement in practice. By construction, RBMs at higher and higher depths in the stack –having fewer and fewer hidden units– represent the data in an increasingly approximate way; their role is to extract more and more compressed encodings of the data distribution and help the bottom RBM sample its energy landscape. The idea that deep representations can be used for better mixing has been previously advocated Bengio et al. (2013) on a strong intuitive basis. To our knowledge the present work provides the first theoretical analysis (in a minimal, overparametrized regime) supporting these intuitions, with a precise quantification of the mixing speed up offered by hierarchical clustering of deeper representations.

## 2 MODELS AND DATASETS

### 2.1 RESTRICTED BOLTZMANN MACHINES

**Definitions.** RBMs are undirected probabilistic models with two layers. A visible layer, which carries data points, is connected to a hidden (representation) layer through a weight matrix $W$, see Fig. 1(a). There are no coupling between units within the same layer. The visible layer includes $N$ units $v_i$, and the hidden layer is made of $M$ units $h_\mu$. For simplicity we hereafter assume $v_i = 0, 1$ (Bernoulli) or $\pm 1$ (similar to spins in statistical physics). The RBM model defines the joint probability

distribution of a visible configuration $\mathbf{v} = \{v_i\}_{i=1\ldots N}$ and a hidden configuration $\mathbf{h} = \{h_\mu\}_{\mu=1\ldots M}$:

$$P(\mathbf{v}, \mathbf{h}) = \frac{1}{Z}\, e^{-E(\mathbf{v}, \mathbf{h})} \quad \text{with} \quad E(\mathbf{v}, \mathbf{h}) = -\sum_{i=1}^{N}\sum_{\mu=1}^{M} W_{i\mu} v_i h_\mu - \sum_{i=1}^{N} g_i\, v_i - \sum_{\mu=1}^{M} c_\mu\, h_\mu \,, \quad (1)$$

where $Z$ is a normalization constant known in physics as the partition function, and the parameters $c_\mu$ and $g_i$ represent biases acting on, respectively, units $h_\mu$ and $v_i$. The probability distribution $P(\mathbf{v})$ of a visible configuration $\mathbf{v}$ can then be computed by marginalizing over the hidden-unit configurations:

$$P(\mathbf{v}) = \sum_{\mathbf{h}} P(\mathbf{v}, \mathbf{h}) = \frac{1}{Z^v}\, e^{-E^v(\mathbf{v})} \,, \quad (2)$$

where $E^v(\mathbf{v})$ is an effective energy function (see Appendix for an explicit expression).

**Alternating Gibbs sampling.** The bipartite nature of the RBM interaction graph suggests a simple sampling algorithm, called Alternating Gibbs Sampling (AGS) and depicted in Fig. 1(b). The basic observation underlying AGS is that the conditional distribution $P(\mathbf{h}|\mathbf{v})$ (respectively $P(\mathbf{v}|\mathbf{h})$) can be factorized over the hidden (respectively, visible) units. AGS consists of sampling from these two distributions in an alternated manner, for a large enough number of steps ensuring convergence to the correct equilibrium distribution $P(\mathbf{v}, \mathbf{h})$). It is important to stress that the simplicity of AGS does not imply it is efficiently sampling the visible and hidden configuration spaces. AGS is generally not more efficient than standard Metropolis sampling of $P(\mathbf{v})$ in the visible space, and is unable to sample distributions with multiple modes separated by large energy barriers (Roussel et al., 2021).

**Training.** The RBM is parametrized by its weights $W = \{w_{i\mu}\}$, as well as the biases $c_\mu$ and $g_i$ considered. All these parameters, generically denoted by $\Theta$ must be learned from the training data, consisting of a set of $K$ samples $\{\mathbf{v}^k\}_{k=1\ldots K}$. This is done through gradient ascent of the log-likelihood of the data, $LL(\Theta) = \frac{1}{K}\sum_{k=1}^{K} \log P(\mathbf{v}^k)$. The generic expression for the gradients is

$$\frac{\partial LL}{\partial \Theta} = -\left\langle \frac{\partial E(\mathbf{v})}{\partial \Theta} \right\rangle_{\text{data}} + \left\langle \frac{\partial E(\mathbf{v})}{\partial \Theta} \right\rangle_{\text{model}} \,, \quad (3)$$

where $\langle . \rangle_{\text{data}}$ denotes the expected value over the data $\{\mathbf{v}^k\}_{k=1\ldots K}$ and $\langle . \rangle_{\text{model}}$ over the model distribution $P(\mathbf{v})$. The latter is an average over an exponential number of terms and is thus untractable, and is estimated in practice with the Persistent Contrastive Divergence algorithm Tieleman (2008). Note that there is no inconsistency between the capability of training a RBM, which directly relies on data to initialize the parallel dynamical chains in persistent constrative divergence, and the hardness of sampling it at later times, by exploring all the modes of the RBM-model distribution. We include regularization in the training procedure, *e.g.*, by replacing $LL(\Theta) \rightarrow LL(\Theta) - \gamma||\Theta||^2$ in the case of a $L_2$ penalty over the model parameters.

## 2.2 DATASETS

We consider several datasets to illustrate the performance of our sampling procedure:

**MNIST.** is a large dataset of $28 \times 28$ pixel images of handwritten digits (LeCun, 1998), see Fig. 1(c). We binarize pixels (white or black). It is empirically known that transitions between very different digits, e.g. 0 and 1, are rarely observed when sampling a RBM trained on MNIST with AGS Béreux et al. (2023); Fernandez-de Cossio-Diaz et al. (2023); Roussel et al. (2021). We test ST on MNIST and MNIST0/1, the restriction to 0 & 1 digits only.

**Lattice Proteins (LP).** are synthetic protein models introduced to investigate protein design, that is, the search for sequences of amino acids likely to fold into a prescribed 3D structure. In the LP models introduced by Shakhnovich & Gutin (1990) and Mirny & Shakhnovich (2001) 27-long sequences $\mathbf{v}$ fold onto $3 \times 3 \times 3$ lattice cubes. The probability $P_{\text{nat}}(S|\mathbf{v})$ that a sequence $\mathbf{v}$ folds into one of the $\simeq 10^5$ distinct structures $S$ is an intricate function of the interactions between nearby amino acids on the structure, see Appendix for details. We choose two widely apart structures $S_A$ and $S_B$ considered by Jacquin et al. (2016), see Fig. 1(d). Our dataset is made of sequences $\mathbf{v}$ with high probabilities to fold in $S_A$ or in $S_B$, *i.e.* such that $P_{\text{nat}}(S_A|\mathbf{v}) > 0.99$ or $P_{\text{nat}}(S_B|\mathbf{v}) > 0.99$.

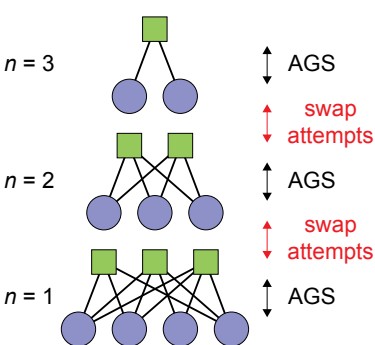

Figure 2: **Principle of Stacked Tempering**, illustrated with a stack of three RBMs. Data are learnt and generated by the bottom RBM. Alternating Gibbs Sampling is used to generate visible and hidden configurations in each RBM. At any step $t$ the configurations $\mathbf{v}_{n+1}^t$ on the visible layer of the $(n+1)^{th}$ RBM and $\mathbf{h}_n^t$ on the hidden layer of the $n^{th}$ RBM can be exchanged, with probability $A_n(\mathbf{h}_n^t, \mathbf{v}_{n+1}^t)$, see Eq. 4.

**2D Ising model.** We consider a celebrated model of magnetism, the 2-dimensional Ising model on a $32 \times 32$ grid with homogeneous interactions between closest neighbors Onsager (1944). This model is hard to sample close to the phase transition temperature, below which the magnetization is non zero. See Appendix C for more details.

## 3 STACKED TEMPERING: PROCEDURE AND APPLICATIONS

We now present the Stacked Tempering (ST) procedure, in which a nested set of RBMs recursively extract relevant collective modes of their predecessors, and, in turn, ease their sampling.

### 3.1 ARCHITECTURE AND TRAINING

The architecture supporting ST is sketched in Fig. 2. It consists of a stack of RBMs, whose widths are such that the number $M_n$ of hidden units of the $n^{th}$ RBM equals the number $N_n$ of visible units of the $(n+1)^{th}$ one. This constraint is essential to make communication between RBM possible, as explained below. In addition, we choose the sizes of visible layers to decrease with the index $n$, *i.e.* $N_n > N_{n+1}$, making deeper RBM 'simpler' than their predecessors in the stack.

The main RBM, which is trained through maximum likelihood on the data $\{\mathbf{v}^k\}$ is the bottom one ($n = 1$). After training, a set of hidden representations $\{\mathbf{h}_1^k\}$ of dimension $M_1$ are obtained by sampling the conditional probabilities $P_1\left(\mathbf{h}|\mathbf{v}^k\right)$ for every $k = 1, ..., K$. These hidden representations are then considered as training data configurations $\{\mathbf{v}_2^k = \mathbf{h}_1^k\}$ for the next RBM, $n = 2$. This is possible since the dimension of the visible layer of the second RBM, $N_2$, is equal to $M_1$. After training of the second RBM a set of representations $\{\mathbf{h}_2^k\}$ are drawn, which are then used as 'data' $\{\mathbf{v}_3^k = \mathbf{h}_2^k\}$ for training the third RBM. The process is iterated all the way up to the last RBM. In summary, the RBMs are trained sequentially, starting from the bottom one.

### 3.2 SAMPLING AND EXCHANGES

Each one of the RBM is sampled with AGS, independently of the others (Fig. 2(a)). In addition, after each AGS step $t$, an attempt is made to swap the hidden, $\mathbf{h}_n^t$, and visible, $\mathbf{v}_{n+1}^t$, configurations of, respectively, the $n^{th}$ and $(n+1)^{th}$ RBM, see Fig. 2(a). This swap occurs with probability

$$A_n(\mathbf{h}_n^t, \mathbf{v}_{n+1}^t) \quad = \quad \min\left(1, \frac{P_{n+1}^v(\mathbf{h}_n^t)\, P_n^h(\mathbf{v}_{n+1}^t)}{P_{n+1}^v(\mathbf{v}_{n+1}^t)\, P_n^h(\mathbf{h}_n^t)}\right) \,, \tag{4}$$

where $P_n^h$ and $P_{n+1}^v$ are the marginal distributions of the hidden and visible configurations of the $n^{th}$ and $(n+1)^{th}$ RBM. The definition of $A_n$ ensures that detailed balance is satisfied. Crucially it does not depend on the intractable normalizations ($Z$ factors) of $P_{n+1}^v, P_n^h$, which cancel out.

Due to the decrease of the layer width with depth (Fig. 2(a)) and to weight regularization, each RBM expresses a coarse-grained approximation of the landscape captured by the previous RBM. The deepest RBM can therefore be easily sampled, and in turn help the RBM below to undergo non local moves in its more complex landscape, without getting trapped in valleys.

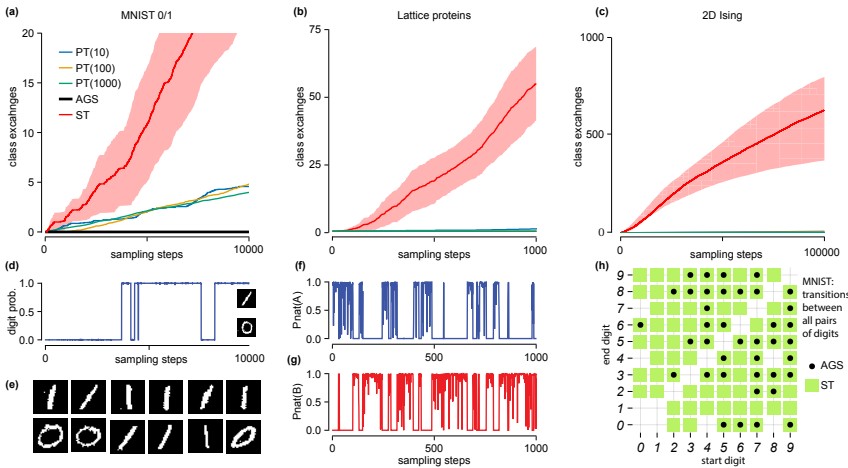

Figure 3: **Application of Stacked Tempering to diverse datasets. (a,b,c)** Cumulative number of transitions between modes during sampling on MNIST0/1, Lattice Proteins, and 2D Ising model, with Stacked Tempering (ST), AGS, and Parallel tempering (PT, legend shows number of intermediate temperatures). Results are averaged over 100 MC chains. For MNIST0/1, a perceptron classifier is used to determine the sampled digits, and transitions between 0 and 1 are recorded. The $x$-axis gives the number of Alternative Gibbs sampling (AGS) steps. Note that different algorithms perform different numbers of AGS per iteration, *e.g.*, PT performs one AGS per intermediate temperature, and ST performs one AGS per stacked RBM. Autocorrelation times are reported in Table S5. **(d)** shows a representative time-course of the classifier score during sampling, while **(e)** shows example sampled digits with AGS (top row) and with ST (below). Digits are displayed every 750 steps and the two dynamics are initialized with the same configuration. ST generates high-quality digits and mixes well between the two classes. **(f,g)** Representative $P_{\text{nat}}(\mathbf{v}|S)$ of sampled sequences for the two structures $S = S_A, S_B$, with ST. ST generates high-quality sequences, folding either on $S_A$ or $S_B$ ($P_{\text{nat}} > 0.9$) for over $80\%$ of the samples, and mixes well between the two families of sequences. **(h)** For RBMs trained on full MNIST, we count the transitions between each pair of digits, using a deep convolutional neural network classifier LeCun et al. (1998). The matrix indicates digit pairs for which at least one transition is observed in average every $10^5$ MC steps for ST (green) and AGS (black). ST is able to jump quicker between many more pairs of digits than AGS.

## 3.3 APPLICATION TO MNIST0/1, MNIST, LATTICE PROTEINS, AND 2D ISING MODEL

We now illustrate the efficiency of ST with respect to AGS and Parallel Tempering (PT). In PT the weights of the RBM are down-scaled by a parameter (called temperature) resulting in a smoother energy landscape. A series of RBMs at different temperatures are allowed to exchange configurations to accelerate sampling. Our implemention follows Desjardins et al. (2010); Cho et al. (2010), with uniform inverse temperatures between 1 (the target distribution) and 0.

**MNIST0/1.** AGS is stuck in one of the two digit classes, see Fig. 3(a). Conversely, ST with a stack of four RBMs ($N_1 = 784$, $M_1 = N_2 = 200$, $M_2 = N_3 = 100$, $M_3 = N_4 = 25$, $M_4 = 10$) is able to frequently jump from one class to the other, see Fig. 3(b). The number of digit switches increases linearly with sampling time, see Fig. 3(c,d). We also implemented PT with increasing number of uniformly distributed intermediate temperatures (d). To compare all algorithms on the same time-scale, the $x$-axis counts the total number of Gibbs sampling steps on any RBM of the stack (PT uses one RBM per temperature, while DT uses one RBM in each layer).

**MNIST.** AGS is able to jump between specific (similar) pairs of digits, such as $0 \rightarrow 5$, or $8 \rightarrow 3$, as shown in Fig. 3h. Thus AGS takes long indirect paths to sample all digit classes. ST, 4 stacked RBMs with $M_1 = 500$, $M_2 = 200$, $M_3 = 50$, and $M_4 = 10$ hidden units, is able to jump faster even many more pairs (even dissimilar) of digits.

**Lattice Proteins.** Three RBMs are used ($N_1 = 27 \times 20$ as amino acids can take 20 possible values, $M_1 = N_2 = 800$, $M_2 = N_3 = 50$, $M_3 = 25$). ST improves mixing between the sequence subspaces associated to structures $S_A$ and $S_B$ and generates high-quality sequences correctly folding either on one or on the other structure, see Fig. 3(e,f). The mixing properties of ST are much better than the

ones of AGS and PT (Fig. 3(g)). AGS and PT almost never switch class (hence the plots overlap around zero).

**2D Ising Model.** A stack of 3 RBMs is used in ST, with 50, 20 and 1 hidden unit. We count the transitions in terms of the magnetization sign. ST performs cluster moves of large groups of spins efficiently, facilitated by the units learning about overall the magnetization of the lattice or patches of nearby spins Mehta & Schwab (2014); Fernandez-de Cossio-Diaz et al. (2023), as is evident from inspection of the weights of the stacked RBMs (see Appendix C 1), whereas AGS and PT keep generating configurations with a fixed magnetization sign (set by the initial configuration). See Fig. S8 for example trajectories of the magnetization during sampling.

The intuitive picture of gradual simplification of landscapes along the stack of RBMs is supported by the applications to MNIST0/1 and Lattice Protein structured data. The RBMs along the stacks progressively cluster the representations of the data, see Appendix Table S1 for the numbers of distinct representations at various depths. Interestingly the cost of introducing additional temperatures in PT is not always offset by accelerated exploration. PT has difficulties in presence of separate modes, a phenomenon akin to critical slowing down in second-order phase transitions, and requires a large number of temperatures near a critical point Katzgraber et al. (2006).

## 4 THEORETICAL ANALYSIS

We now seek to characterize the conditions on the data distribution and on hyperparameters allowing ST to be efficient. To obtain a minimal and analytically tractable framework, we consider the case of strongly overparametrized RBM, *i.e.* with layer widths $N, M \to \infty$. Our analysis is organized along three steps: (1) the determination of the form of the coupling matrix through maximum log-likelihood training; (2) the identification of the pattern separation/pattern completion regime and phase transition depending on how much the RBM is regularized; (3) the calculation of the mixing and swapping times for a stack of two RBMs, and the discussion of when and how much ST accelerates sampling.

### 4.1 A MINIMAL SETTING FOR ANALYZING RBM DATA-TO-REPRESENTATION MAPPING

We consider the minimal case of $K = 2$ data configurations, $\mathbf{v}^1$ and $\mathbf{v}^2$, with dot product $x = \mathbf{v}^1 \cdot \mathbf{v}^2 / N$; Here, visible and hidden units are chosen to take $\pm 1$ values. We learn these two data points with a RBM, through maximization of the regularized log-likelihood,

$$LL = \frac{1}{2} \left[ \log P(\mathbf{v}^1) + \log P(\mathbf{v}^2) \right] - \frac{N\gamma}{2} \sum_{i\mu} W_{i\mu}^2 \tag{5}$$

over the weights $W_{i\mu}$, where $\gamma > 0$ is the regularization strength. As shown in Appendix E, in the limit $N, M \to \infty$ at fixed ratio $\alpha = \frac{M}{N}$, the trained weights are of the form (after a suitable ordering of the hidden units)

$$W_{i\mu} = \frac{1}{N} \times \begin{cases} \frac{v_i^1 - v_i^2}{1-x} W_- & \text{if} \quad 1 \leq \mu \leq \frac{1-y}{2} M , \\ \frac{v_i^1 + v_i^2}{1+x} W_+ & \text{if} \quad \frac{1-y}{2} M < \mu \leq M , \end{cases} \tag{6}$$

where $y = \mathbf{h}^1 \cdot \mathbf{h}^2 / M$ is the overlap between the most likely hidden representations $\mathbf{h}^1, \mathbf{h}^2$ of the data points $\mathbf{v}^1, \mathbf{v}^2$. The values of $W_{\pm}$ are chosen to maximize $LL$, resulting in the stationarity equations:

$$\gamma W_{\pm} = \tanh\left(\frac{1 \pm x}{2} W_{\pm}\right) - m_{\pm} n_{\pm}, \ m_{\pm} = \tanh\left(\frac{1 \pm y}{2} \alpha W_{\pm} n_{\pm}\right), \ n_{\pm} = \tanh\left(W_{\pm} m_{\pm}\right) \tag{7}$$

These $y$-dependent weights can be injected back into the LL in Eq. 5, allowing us to determine the value of $y \in [-1; +1]$ maximizing the log-likelihood, hereafter referred to as $y^*(x, \alpha, \gamma)$.

### 4.2 REPRESENTATIONAL REGIMES AND PHASE TRANSITION

Our first aim is to understand the data-to-representation mapping, more precisely, how the representation similarity $y^*$ is related to the overlap between data points, $x$. We first find that $y^*$ is decreasing with $\alpha$ and increasing with $\gamma$ at fixed data overlap $x$, see Fig. 4(a,b). In other words, regularizing more the RBM, by decreasing either the amplitude of the weights (through higher $L_2$ penalty $\gamma$) or

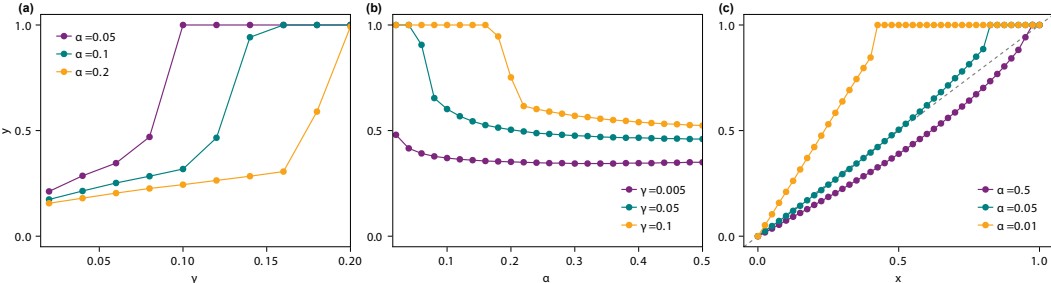

Figure 4: Optimal overlap $y^*(\gamma, x, \alpha)$. (a) Behaviour of $y^*$ vs. $\gamma$ at fixed $x = 0.2$ (b) Behaviour of $y^*$ vs. $\alpha$ at fixed $x = 0.5$. (c) Behaviour of $y$ vs. $x$ at fixed $\gamma = 0.015$; the dashed lines shows $y = x$.

the number of hidden units (equivalently, the aspect ratio $\alpha$), produces more similar representations. For strong enough regularization, we obtain $y^* = 1$, showing that the two representations become identical though the data points are not.

We then plot $y^*$ as a function of $x$ in Fig. 4(c). We identify three different regimes. (i) For low–to–intermediate data overlaps and low regularization (small $\gamma$, large $\alpha$), $y^*$ is a slowly increasing function of $x$, with a slope smaller than unity. The RBM has a tendency to produce representations less correlated than the corresponding data. This regime can be referred to as *pattern separation*, a vocable used in the context of computational neuroscience (Rolls, 2013). (ii) For low–to–intermediate data overlaps and large regularization (large $\gamma$, small $\alpha$), $y^*$ is a quickly increasing function of $x$, with a slope larger than unity. The RBM has a tendency to produce representations more correlated than the data. This regime is reminiscent of *pattern completion* (Rolls, 2013). (iii) For large data overlaps, *i.e.* for $x > x_c$, $y^* = 1$: the RBM has mapped similar but distinct data onto a unique representation. This regime can be referred to as *clustering* and can be seen as an extreme version of pattern completion. The value of $x_c$ as a function of $\alpha, \gamma$ can be computed as discussed in Appendix E 5.

The boundary between the pattern separation and completion regimes is a phase transition, clearly visible in the abrupt change of slope of $y^*$ in Figs. 4(a,b). Details about the phase transition and the attached order parameter can be found in Appendix E. This phase transition has a concrete interpretation in terms of the generative diversity of the RBM. Once the training is done we can use our RBM to generate 'new' data through Alternating Gibbs Sampling. In the *pattern completion* regime, whatever the initial configuration $\mathbf{v} = \mathbf{v}^1$ or $\mathbf{v}^2$ on the visible layer, the RBM will generate the same data distribution. In the *pattern separation* regime, the ergodicity of the visible configuration space is broken, and the RBM will generate configurations $\mathbf{v}$ similar to the data point present at the initialization of the MC chain.

As a final note let us emphasize that the representation regimes and phase transition identified here is not specific to the case $K = 2$. The analytical calculation of the log-likelihood $LL$ done above can be extended to any finite $K > 2$ (while $N, M$ are sent to infinity). Analysis of the extremization equations for $K$ generic data points shows that, as $\gamma$ are progressively decreased, a sequence of clustering-like phase transitions takes place, with more and more similar representations. The number of distinct representations varies from $K$ to 1 as regularization is made stronger.

### 4.3 ANALYTICAL CALCULATION OF THE AVERAGE MIXING AND SWAP TIMES FOR A STACK OF TWO RBMS

We now analytically calculate the speed up offered by ST with respect to AGS in the case of a stack of two RBM learning $K = 2$ data points. The two RBM are defined by aspect ratios $\alpha_1, \alpha_2$ and regularization strengths $\gamma_1, \gamma_2$, see Fig. 5(a). We call $x$ the overlap between the data points. The overlaps $y_1$ between their representations in the first RBM, and $y_2$ between the representations of these representations in the second RBM can be determined as functions of the other parameters according to the previous Sections, with the results $y_1 = y^*(x, \alpha_1, \gamma_1)$ and $y_2 = y^*(y_1, \alpha_2, \gamma_2)$.

**Mixing time with AGS for a single RBM.** We consider a single RBM, with data and representations overlaps equal to, respectively, $x$ and $y$, aspect ratio $\alpha$ and $L_2$ penalty $\gamma$, as in Section 4.1. Extending

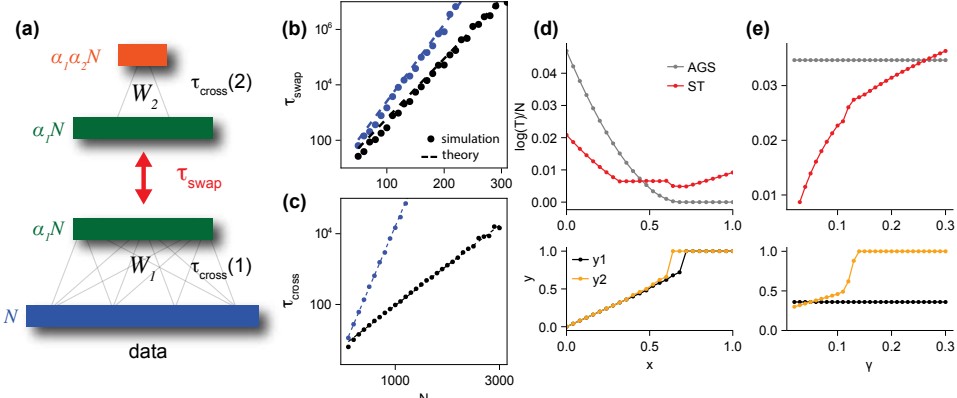

Figure 5: **Theoretical analysis of Stacked Tempering**. **(a).** Stack of two RBMs. $\tau_{\text{cross}}^{(1,2)}$ are the mixing times of the two machines, $\tau_{swap}$ is the average time between swaps of $\mathbf{h}_1$ and $\mathbf{v}_2$. **(b,c)** Exponential scaling of the swap **(b)** and **(c)** mixing times as a function of the data dimension $N$. Two parameter settings (values in Appendix H) are shown in each panel. **(d,e)** Average switching times (in log-scale) for ST and AGS, vs. **(d)** the data overlap $x$ at fixed $w_2 = 5, \alpha_2 = 0.3$, and **(e)** the penalty strength $\gamma$ of the top RBM, at fixed $x = 0.5, \alpha_2 = 0.5$. Bottom panels show $y_2$ maximizing the likelihood of the top RBM.

the statistical mechanics approach that allowed us to determine the value of $LL$ we can compute the free-energy barrier separating (when $y < 1$) the two minima of the energy landscape. This barrier, through Arrhenius law, gives access to the mixing time for switching from one minimum to another, $\tau_{\text{cross}} \sim e^{N\,\mathcal{B}(x,\alpha,\gamma)}$ where the expression the barrier height is

$$\mathcal{B}(x,\alpha,\gamma) = \underset{m_-,n_-}{\text{extr}} \left\{ \alpha \left( \frac{1-y}{2} \right) W_- m_- n_- + \frac{1-x}{2} \mathcal{H}(m_-) + \alpha \left( \frac{1-y}{2} \right) \mathcal{H}(n_-) \right\} \quad (8)$$

with $y = y^*(x,\alpha,\gamma)$, and where $\mathcal{H}(m)$ is the binary entropy function. Detailed calculation can be found in Appendix G. Note that the time $\tau_{\text{cross}}$ is exponential in $N$, showing that mixing is extremely slow, as expected for AGS of a multimodal data distribution (Roussel et al., 2021). This asymptotic result is in very good agreement with numerical estimates for finite $N$, see Fig. 5(c).

The result above can be used to estimate the crossing times $\tau_{\text{cross}}^{(1)}$ and $\tau_{\text{cross}}^{(2)}$ of the two RBMs (when they are not coupled through replica exchange), with appropriate choices of the parameters.

**Swap time between two RBMs.** We now study how the two RBMs in Fig. 5(a), which generate, respectively, configurations $\{\mathbf{v}_1^t, \mathbf{h}_1^t\}$ and $\{\mathbf{v}_2^t, \mathbf{h}_2^t\}$, can occasionally exchange their configurations. The swap between $\mathbf{h}_1^t$ and $\mathbf{v}_2^t$ is accepted with probability $A_1(\mathbf{h}_1^t, \mathbf{v}_2^t)$, see Eq. 4. The mean value $\langle A \rangle$ of this acceptance ratio can be computed, and its inverse is an estimate of the characteristic time $\tau_{\text{swap}}$ between two replica exchanges (see Appendix F for a detailed calculation), $\tau_{\text{swap}} = \frac{1}{\langle A \rangle} \sim e^{\alpha_1 N (G^* - G^0)}$, to leading order in $N$, where

$$G^* = \min_{m_+,m_-} G(m_+,m_-,m_+,m_-) \geq G^0 = \min_{m_+,m_-,n_+,n_-} G(m_+,m_-,n_+,n_-) \quad (9)$$

and $G(m_+,m_-,n_+,n_-)$ is a free energy cost associated to swapping the configurations of the neighboring RBMs:

$$G(m_+,m_-,n_+,n_-) = E_1^h(n_+,n_-) - \mathcal{S}(n_+,n_-) + E_2^v(m_+,m_-) - \mathcal{S}(m_+,m_-) \quad (10)$$

where $E_1^h, E_2^v$ are the effective energies of the two RBMs at the contact layer, and $\mathcal{S}(t_+,t_-) = \frac{1+y}{2}\mathcal{H}(t_+) + \frac{1-y}{2}\mathcal{H}(t_-)$ is an entropic contribution. This analytical prediction for $\tau_{\text{swap}}$ is in excellent agreement with numerical estimates, see Fig. 5(b).

**Speed-up of ST with respect to AGS.** We now combine the results above to determine the gain offered by ST over AGS. Sampling the bottom RBM (parameters $\alpha_1, \gamma_1$) with AGS will lead to a switching time between modes of the order of $\tau_{\text{AGS}} \equiv \tau_{\text{cross}}^{(1)} \sim e^{N\,\mathcal{B}(x,\alpha_1,\gamma_1)}$. By comparison,

sampling with ST will take time

$$\tau_{\text{ST}} = \max\left(\tau_{\text{cross}}^{(2)}, \tau_{\text{swap}}\right) \sim e^{N \max\left(\mathcal{B}(y_1, \alpha_2, \gamma_2), \alpha_1(G^* - G^0)\right)} . \tag{11}$$

Results for $\tau_{\text{ST}}$ and $\tau_{\text{AGS}}$ are shown in Fig. 5(d,e) for representative parameter values of $\alpha_2$ and $\gamma_2$. We observe the existence of an optimal speed up with ST, well below the value of the time required with AGS. This optimum implements a trade-off between strong compression of the representations - low acceptance rate of the swap attemps (obtained for instance for small $\alpha_2$ or large $\gamma_2$ for the top RBM) and weak compression - high acceptance (for large $\alpha_2$, small $\gamma_2$).

## 5 DISCUSSION

In this work, we have first introduced a sampling algorithm for RBM, building upon the ideas of Parallel and Deep Tempering. Contrary to the latter Desjardins et al. (2014), our Stacked Tempering (ST) procedure is not a training algorithm for DBNs. By decreasing the number of hidden units with the depth along the stack, the RBMs encode increasingly simplified versions of the bottom energy landscape of interest. The fast mixing of high RBMs then propagates down through replica exchanges. We have shown, on varied datasets coming from machine learning, computational biology and statistical physics, that ST is more efficient than Alternate Gibbs Sampling and Parallel Tempering, even with a much larger number of parallel chains.

In the second part of the paper we have provided analytical support to the observed gain in mixing time offered by ST, based on careful analysis of a stack with two RBMs. To obtain an analytically tractable framework, we have considered the case of largely overparametrized RBMs. This setup allowed us to determine a transition between a pattern separation and a pattern completion (clustering) regimes for the hidden representations, controlled by the hyperparameters of the RBMs. This transition has profound impact on the average mixing and swapping times, emphasizing how the mapping between data and representations and the efficiency of sampling with ST are intimately interconnected. Let us also stress that, despite the minimal nature of this mathematical setting, we were able to obtain non-trivial analytical results in a field where exactly solvable models are not frequent. We also observe analogous clustering and separation regimes in real datasets (see Fig. S5, S6, S4).

An apparent limitation of our sampling procedure is the cost of training the stack of RBMs compared to other methods. In the architectures we consider, deeper RBMs have however less units and less rugged landscapes, and we find that their training converges faster (see Tables S3 and S4). Thus in practice, this additional training cost might not be significant. Another limitation of our work is that our theoretical analysis focuses on a over-parameterized regime with few data points. This setting allowed us to derive exact analytical expressions that we believe yield insight into more general situations, but it remains unclear how it could be extended to other settings, for instance, when the number $K$ of data points is comparable to their dimension $N$.

Our work could be extended in many directions. First, we could exploit the benefit of most efficient mixing provided by ST during the learning phase itself. Exchange efficiency across the stack could be used as a secondary training objective, *e.g.* through terms such as $\log A$ between contiguous RBMs. These contributions, in addition to the log-likelihoods already considered in the training, would favor a smoother energy landscape at the bottom of the stack, in a manner akin to regularization. More generally, one would like to have a better understanding of how to optimize hyper-parameters of the stack (number of RBM layers, hidden units, regularization). Second, a conceptually appealing feature of ST is that each single data point is associated to a sequence of representations, one per RBM in the stack. It is tempting to associate these multi-representations to multiple levels of adaptive coarse-graining of the data, built by the RBMs in a fully unsupervised way. A better characterization of these multi-representations would be very interesting. From this point of view, we believe that our ST procedure, which lies half way between a single RBM and a full DBN could offer a valuable path to better investigate the properties of deep unsupervised networks. Last of all, ST could be applied to model physical systems, where it is desired to approximate or sample a known classical or quantum Hamiltonian Carleo et al. (2019); Carleo & Troyer (2017), or to aid training deep models like DBN.

**Acknowledgments.** We thank Alessio Giorlandino for useful discussions. JFdCD acknowledges support from Université PSL AI Junior Fellow program. JFdCD, CR, SC, RM acknowledge support from Grant No. ANR-19 Decrypted CE30-0021-01.

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
