# Accelerated Sampling with stacked Restricted Boltzmann Machines

**SUPPLEMENTARY MATERIAL**

Jorge Fernandez-de-Cossio-Diaz*, Clément Roussel*, Simona Cocco, Rémi Monasson

(∗) equal contribution

## Appendix A: Restricted Boltzmann machines

### 1. Marginal distributions over visible and hidden states

As we saw in the main text, the RBM defines the following marginal distribution over visible configurations given by:

$$P(\mathbf{v}) = \sum_{\mathbf{h}} P(\mathbf{v}, \mathbf{h}) = \frac{1}{Z^v} e^{-E^v(\mathbf{v})} , \tag{A1}$$

where the effective energy, $E^v(\mathbf{v})$, accounts for effective interactions that arise among visible units on account of the marginalized hidden variables:

$$E^v(\mathbf{v}) = -\ln \sum_{\mathbf{h}} \exp \left( \sum_{i=1}^{N} \sum_{\mu=1}^{M} W_{i\mu} v_i h_\mu + \sum_{i=1}^{N} g_i v_i + \sum_{\mu=1}^{M} c_\mu h_\mu \right) \tag{A2}$$

Taking advantage of the fact that visible units do not interact with hidden units, we can evaluate $E^v(\mathbf{v})$, as follows:

$$E^v(\mathbf{v}) = -\sum_{i=1}^{N} g_i v_i - \sum_{\mu=1}^{M} \Gamma_\mu(I_\mu(\mathbf{v})) \tag{A3}$$

Here, $I_\mu(\mathbf{v}) = \sum_{i=1}^{N} W_{i\mu} v_i$ denotes the input received by hidden unit $h_\mu$ and $\Gamma_\mu(I) = \log \sum_h \exp\left[h\left(c_\mu + I_\mu(\mathbf{v})\right)\right]$ is simple to compute for binary units $h$.

A similar expression is obtained for the effective energy $E^h(\mathbf{h})$ corresponding to the log-probability of hidden configurations, obtained through marginalization over visible configurations.

### 2. Alternative Gibbs sampling

The alternative Gibbs sampling procedure is defined as follows:

- Starting from a visible configuration $\mathbf{v}^t$ at step $t$, a hidden configuration $\mathbf{h}^{t+1}$ is drawn from $P(\mathbf{h}|\mathbf{v}^t) = \prod_{\mu=1}^{M} P(h_\mu|\mathbf{v}^t)$. Here $P(h_\mu|\mathbf{v}) \propto \exp\left[h_\mu(c_\mu + I_\mu(\mathbf{v}))\right]$. This step can be seen as a stochastic extraction of features from the configuration $\mathbf{v}^t$.

- A new visible configuration $\mathbf{v}^{t+1}$ is drawn from $P\left(\mathbf{v}|\mathbf{h}^{t+1}\right) = \prod_{i=1}^{N} P(v_i|\mathbf{h}^{t+1})$. Here $P(v_i|\mathbf{h}) \propto \exp\left[v_i(g_i + I_i(\mathbf{h}))\right]$, where $I_i(\mathbf{h})$ is the input of the visible unit $v_i$, i.e $I_i(\mathbf{h}) = \sum_{\mu=1}^{M} W_{i\mu} h_\mu$. This step can be seen as a stochastic reconstruction of $\mathbf{v}$ from the representation $\mathbf{h}^{t+1}$.

Given a large number of iterations, it is guaranteed to converge to the correct marginal distributions.

### 3. Training

The RBMs are trined by persistent contrastive divergence [1]. All RBMs are trained for 50000 stochastic gradient descent iterations (regardless of the size of the dataset) using Adam [2]. Batchsize was fixed at 128, and the persistent chains are updated with 20 Monte Carlo steps per iteration. The implementation follows closely previous works [3].

## Appendix B: Lattice proteins

The lattice cube defines a set of $\mathcal{N} = 103,406$ distinct structures. The probability of a sequence $\mathbf{v}$ to fold in a given structure $S$ is expressed as

$$P_{\mathrm{nat}}(S|\mathbf{v}) = \frac{\exp\left(-E_{LP}(\mathbf{v}, S)\right)}{\sum_{S'} \exp\left(-E_{LP}(\mathbf{v}, S')\right)} \tag{B1}$$

where the sum runs over all $\mathcal{N}$ possible structures $S'$. For a given structure $S$, there are 28 contacts between the amino acids, which in turn determine the energy of the sequence $\mathbf{v}$ in a structure $S$, as:

$$E_{LP}(\mathbf{v}, S) = \sum_{i<j} c_{ij}^S \, \Delta E_{MJ}(v_i, v_j) \tag{B2}$$

where $c_{ij}^S = 1$ if sites $i$ and $j$ are in contact in the fold $S$; otherwise, $c_{ij}^S = 0$. The pairwise energy $\Delta E_{MJ}(v_i, v_j)$ is the Miyazawa-Jernigan (MJ) potential, a proxy for physico-chemical interactions between nearby amino acids [4].

## Appendix C: Two-dimensional Ising model

We next consider the standard Ising model [5] on a two-dimensional regular $L \times L$ square grid, with $L = 32$, with uniform positive interactions between nearest-neighbor spins. The values of the interaction, or, equivalently, of the inverse temperature, can be varied to explore both paramagnetic (weak interactions) and ferromagnetic (strong interactions) regimes. As is well known, the model suffers a phase transition from a disordered state to an ordered one, at an inverse temperature $\beta_c \approx 0.44$.

The RBM is trained on a large number of equilibrium configurations of the Ising model, which are obtained by a combination of Wolff cluster algorithm and local Metropolis sampling steps. The implementation is the same as in [3].

### 1. Representations at different levels of the stacked RBMs on the Ising model

To interpret the weights of the stacked RBMs we compute the "effective" weights of a hidden unit in a top layer when projected on the bottom layers. More precisely, we define effective weights $\hat{W}$ through the recurrence relation:

$$\hat{W}_{ik}^\ell = \sum_j W_{ij}^{\ell-1} W_{jk}^\ell \tag{C1}$$

that for the bottom RBM is initialized simply as $\hat{W}_{ij}^1 = W_{ij}^1$. A similar definition has been used by [6] in connection to a renormalization group interpretation of stacked RBMs. These effective weights are an approximate representation of the indirect effect a hidden unit of an RBM along the stack has on the bottom Ising lattice. Figure S1 plots these representations for selected hidden units across the stack. As claimed in the main text, the hidden units at different level capture information about the magnetization of localized patches in the Ising lattice. In particular, the hidden unit at the top RBM has all positive effective weights (between $\approx 3$ and $\approx 7$), and therefore computes an approximate version of the total magnetization of the lattice. Indeed, if we sample hidden units progressively bottom to top, starting from any equilibrium configuration of the Ising model at the bottom, we find that for $> 98\%$ of cases the top hidden unit encodes for the sign of the total magnetization of the Ising configuration

## Appendix D: Numbers of distinct representations at various depths in the stack for MNIST0/1 and Lattice Protein datasets

Like many real datasets, MNIST0/1 and Lattice Proteins data are not simply organized in hierarchical and regular clusters. Nevertheless, the different RBM at different levels in the stack, progressively cluster the representations of the data. We computed the numbers of distinct representations at various depths, by finding the minima of $E^h(\mathbf{h})$ for each RBM. To do this, we start by random configurations $\mathbf{h}$ that we then greedily optimize until a local minimum of $E^h(\mathbf{h})$ is found. The results are shown in Table S1.

TABLE S1. Number of distinct representations of the data in the hidden layers, for the 4 stacked RBM in the MNIST0/1 dataset and for the 3 stacked RBM for Lattice Proteins $S_A/S_B$.

| Level | n=1 | n=2 | n=3 | n=4 |
|---|---|---|---|---|
| MNIST | 12635 | 12560 | 3099 | 68 |
| Lattice proteins | 96234 | 39345 | 1273 | - |

We also computed the products $w\alpha$ that appear in the theory for the RBMs trained on MNIST0/1 and Lattice proteins. See Table S2.

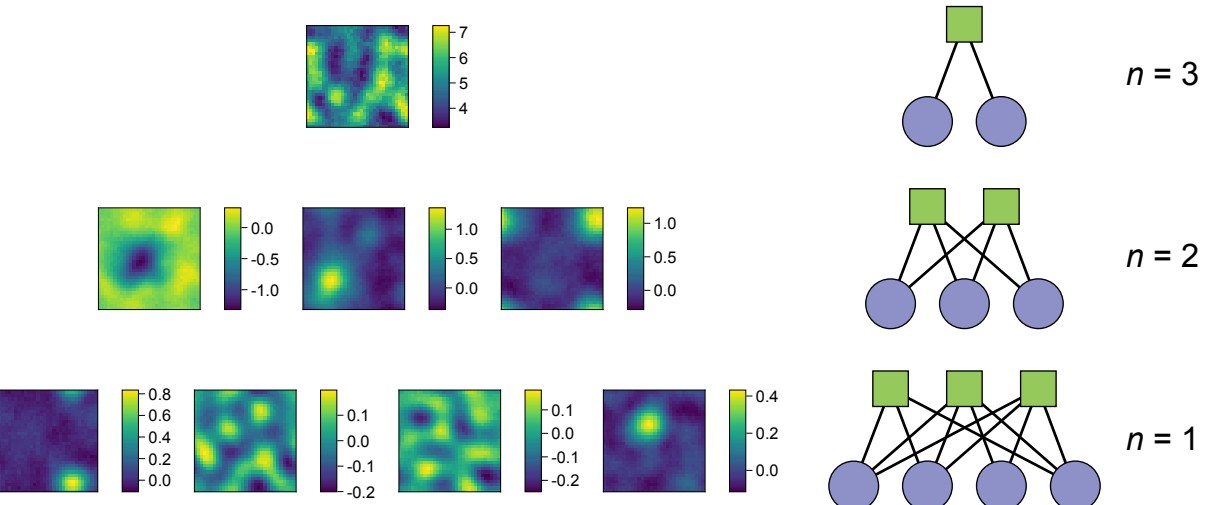

FIG. S1. Representations of the weights of selected hidden units at each level of the stack of 3 RBMs trained on the 2-dimensional Ising model. For each RBM at each level, we represent the effective weights (defined in Eq. (C1)) of selected hidden units.

TABLE S2. Values of the products $w\alpha$ for the RBMs trained on MNIST0/1 and for Lattice Proteins $S_A/S_B$.

| Level | n=1 | n=2 | n=3 | n=4 |
|---|---|---|---|---|
| MNIST | 2.01 | 2.96 | 2.31 | 1.69 |
| Lattice proteins | 38.80 | 0.88 | 2.37 | - |

## Appendix E: Solution of the RBM trained on two data points

In this section we discuss the weights of a spin RBM trained on two data points, justifying Equation (6) from the main text, that we rewrite here

$$W_{i\mu} = \frac{1}{N} \times \begin{cases} \frac{v_i^1 + v_i^2}{1+x} W_+ & \text{if } \mu > \psi M \\ \frac{v_i^1 - v_i^2}{1+x} W_- & \text{if } \mu < \psi M \end{cases} \tag{E1}$$

That is, the hidden units of the RBM split into two groups, with a first group having weights $w_{i\mu} \propto v_i^1 + v_i^2$ and a second group having weights $w_{i\mu} \propto v_i^1 - v_i^2$. Here $\psi = (1-y)/2$ and $1-\psi$ are the fractions of hidden units of the two kinds. In comparison to Eq. (6) in the main text, note that we here included the factors $2/(1 \pm x)$, added here for convenience.

It is also convenient to define the vectors along which the weights align,

$$U_i^\pm = \frac{1}{2}(v_i^1 \pm v_i^2) \tag{E2}$$

giving respectively the center of mass of the two data points ($U_i^+$) and their separation ($U_i^-$).

In this section, we will show that the weights (E1) give a stationary point of the regularized log-likelihood that is also a stable maximum. To do this, we consider a small perturbation $\epsilon_{i\mu}$ added to the weights, (E1),

$$\tilde{w}_{i\mu} = w_{i\mu} + \epsilon_{i\mu} \tag{E3}$$

We then Taylor-expand the regularized log-likelihood,

$$\tilde{\mathcal{L}} = -\langle \tilde{E}^v(\mathbf{v}) \rangle_{\mathcal{D}} - \ln \tilde{Z} - \frac{N\gamma}{2} \sum_{i\mu} \tilde{w}_{i\mu}^2 \tag{E4}$$

in powers of $\epsilon_{i\mu}$ up to second-order (hereafter a tilde denotes a quantity evaluated at the perturbed weights $\tilde{w}_{i\mu}$).

Expanding each of the terms in (E4) results in:

$$-\tilde{E}^v(\mathbf{v}) \approx -E^v(\mathbf{v}) + \sum_{i\mu} \epsilon_{i\mu} v_i \langle h_\mu | \mathbf{v} \rangle + \sum_{ij} \sum_\mu \frac{\epsilon_{i\mu} \epsilon_{j\mu}}{2} v_i v_j (1 - \langle h_\mu | \mathbf{v} \rangle^2)$$

$$\ln \tilde{Z} \approx \ln Z + \sum_{i\mu} \epsilon_{i\mu} \langle v_i h_\mu \rangle + \frac{1}{2} \sum_{ij} \sum_{\mu\nu} \epsilon_{i\mu} \epsilon_{j\nu} \left( \langle v_i v_j h_\mu h_\nu \rangle - \langle v_i h_\mu \rangle \langle v_j h_\nu \rangle \right) \tag{E5}$$

$$\sum_{i\mu} \tilde{w}_{i\mu}^2 \approx \sum_{i\mu} w_{i\mu}^2 + 2 \sum_{i\mu} w_{i\mu} \epsilon_{i\mu} + \sum_{i\mu} \epsilon_{i\mu}^2$$

where the moments are evaluated under the unterturbed RBM with weights (E1). The effective energy is then to be averaged over the two data points, $\mathbf{v}^{1,2}$, giving

$$-\langle \tilde{E}^v(\mathbf{v}) \rangle_{\mathcal{D}} \approx -\langle E^v(\mathbf{v}) \rangle_{\mathcal{D}} + \tanh(W_+) \sum_i \sum_{\mu > M\psi} \epsilon_{i\mu} U_i^+ + \tanh(W_-) \sum_i \sum_{\mu < M\psi} \epsilon_{i\mu} U_i^-$$

$$+ \sum_{ij} (U_i^+ U_j^+ + U_i^- U_j^-) \left( \sum_\mu \frac{\epsilon_{i\mu} \epsilon_{j\mu}}{2} - \tanh^2(W_+) \sum_{\mu > M\psi} \frac{\epsilon_{i\mu} \epsilon_{j\mu}}{2} - \tanh^2(W_-) \sum_{\mu < M\psi} \frac{\epsilon_{i\mu} \epsilon_{j\mu}}{2} \right) \tag{E6}$$

The second and fourth-order moments of the RBM, appearing in the expansion of $\ln Z$, are also easy to evaluate under the weights (E1). After some algebra, we find that, in the large $N$ limit:

$$\ln \tilde{Z} \approx \ln Z + \sum_{i\mu} \epsilon_{i\mu} \langle v_i h_\mu \rangle + \sum_{ij} \sum_{\mu\nu} \frac{\epsilon_{i\mu} \epsilon_{j\nu}}{2} \left( \langle v_i v_j h_\mu h_\nu \rangle - \langle v_i h_\mu \rangle \langle v_j h_\nu \rangle \right)$$

$$= \ln Z + m_+ n_+ \sum_i \sum_{\mu > M\psi} \epsilon_{i\mu} U_i^+ + m_- n_- \sum_i \sum_{\mu < M\psi} \epsilon_{i\mu} U_i^-$$

$$+ \left( 1 - \frac{1+x}{2} m_+^2 - \frac{1-x}{2} m_-^2 \right) \sum_i \left( n_+^2 \sum_{\mu,\nu > M\psi} \frac{\epsilon_{i\mu} \epsilon_{i\nu}}{2} + n_-^2 \sum_{\mu,\nu < M\psi} \frac{\epsilon_{i\mu} \epsilon_{i\nu}}{2} \right)$$

$$+ m_-^2 n_+^2 \sum_{ij} \sum_{\mu,\nu > M\psi} \frac{\epsilon_{i\mu} \epsilon_{j\nu}}{2} U_i^- U_j^- + m_+^2 n_-^2 \sum_{ij} \sum_{\mu,\nu < M\psi} \frac{\epsilon_{i\mu} \epsilon_{j\nu}}{2} U_i^+ U_j^+$$

$$+ m_+ m_- n_+ n_- \sum_{ij} \left( U_i^- U_j^+ \sum_{\mu > M\psi > \nu} \frac{\epsilon_{i\mu} \epsilon_{j\nu}}{2} + U_i^+ U_j^- \sum_{\mu < M\psi < \nu} \frac{\epsilon_{i\mu} \epsilon_{j\nu}}{2} \right) \tag{E7}$$

where the quantities $m_\pm, n_\pm$ are "order parameters" that determine the magnetizations of the visible and hidden units of the RBM, and satisfy the saddle-point equations (also given in the main text at Eq. (6))

$$m_\pm = \tanh \left( \frac{2\alpha_\pm W_\pm n_\pm}{1 \pm x} \right)$$

$$n_\pm = \tanh (W_\pm m_\pm) \tag{E8}$$

where $\alpha_+ = (1 - \psi)\alpha$ and $\alpha_- = \psi\alpha$.

Finally for the regularization term, we may rewrite it as:

$$\frac{N\gamma}{2} \sum_{i\mu} \tilde{w}_{i\mu}^2 = \frac{N\gamma}{2} \left( \sum_{i\mu} w_{i\mu}^2 + 2 \sum_{i\mu} w_{i\mu} \epsilon_{i\mu} + \sum_{i\mu} \epsilon_{i\mu}^2 \right)$$

$$= \frac{N\gamma}{2} \sum_{i\mu} w_{i\mu}^2 + \frac{2W_+ \gamma}{1+x} \sum_i \sum_{\mu > M\psi} U_i^+ \epsilon_{i\mu} + \frac{2W_- \gamma}{1-x} \sum_i \sum_{\mu < M\psi} U_i^- \epsilon_{i\mu} + \frac{N\gamma}{2} \sum_{i\mu} \epsilon_{i\mu}^2 \tag{E9}$$

Substituting everything back into (E4), gives the desired second-order expansion:

$$
\begin{aligned}
\tilde{\mathcal{L}} &= -\langle \tilde{E}_{\text{eff}}(\mathbf{v}) \rangle_{\mathcal{D}} - \ln \tilde{Z} - \frac{N\gamma}{2} \sum_{i\mu} \tilde{w}_{i\mu}^2 \\
&= \mathcal{L} + \tanh(W_+) \sum_i \sum_{\mu > M\psi} \epsilon_{i\mu} U_i^+ + \tanh(W_-) \sum_i \sum_{\mu < M\psi} \epsilon_{i\mu} U_i^- \\
&\quad + \sum_{ij} (U_i^+ U_j^+ + U_i^- U_j^-) \left( \sum_\mu \frac{\epsilon_{i\mu} \epsilon_{j\mu}}{2} - \tanh^2(W_+) \sum_{\mu > M\psi} \frac{\epsilon_{i\mu} \epsilon_{j\mu}}{2} - \tanh^2(W_-) \sum_{\mu < M\psi} \frac{\epsilon_{i\mu} \epsilon_{j\mu}}{2} \right) \\
&\quad - m_+ n_+ \sum_i \sum_{\mu > M\psi} \epsilon_{i\mu} U_i^+ - m_- n_- \sum_i \sum_{\mu < M\psi} \epsilon_{i\mu} U_i^- \\
&\quad - \left( 1 - \frac{1+x}{2} m_+^2 - \frac{1-x}{2} m_-^2 \right) \sum_i \left( n_+^2 \sum_{\mu,\nu > M\psi} \frac{\epsilon_{i\mu} \epsilon_{i\nu}}{2} + n_-^2 \sum_{\mu,\nu < M\psi} \frac{\epsilon_{i\mu} \epsilon_{i\nu}}{2} \right) \\
&\quad - m_-^2 n_+^2 \sum_{ij} \sum_{\mu,\nu > M\psi} \frac{\epsilon_{i\mu} \epsilon_{j\nu}}{2} U_i^- U_j^- - m_+^2 n_-^2 \sum_{ij} \sum_{\mu,\nu < M\psi} \frac{\epsilon_{i\mu} \epsilon_{j\nu}}{2} U_i^+ U_j^+ \\
&\quad - m_+ m_- n_+ n_- \sum_{ij} \left( U_i^- U_j^+ \sum_{\mu > M\psi > \nu} \frac{\epsilon_{i\mu} \epsilon_{j\nu}}{2} + U_i^+ U_j^- \sum_{\mu < M\psi < \nu} \frac{\epsilon_{i\mu} \epsilon_{j\nu}}{2} \right) \\
&\quad - \frac{2W_+ \gamma}{1+x} \sum_i \sum_{\mu > M\psi} U_i^+ \epsilon_{i\mu} - \frac{2W_- \gamma}{1-x} \sum_i \sum_{\mu < M\psi} U_i^- \epsilon_{i\mu} - \frac{N\gamma}{2} \sum_{i\mu} \epsilon_{i\mu}^2
\end{aligned}
\tag{E10}
$$

Assuming that the weights satisfy the stationarity equation given in the main-text,

$$
\gamma W_\pm = \tanh\left( \frac{1 \pm x}{2} W_\pm \right) - m_\pm n_\pm
\tag{E11}
$$

which can be found by differentiation of the log-likelihood with respect to $W_\pm$, we find that all the terms of linear-order in (E10) cancel, confirming the stationarity of the weights (E1).

To study the stability of this critical point of the log-likelihood, we must consider the second-order terms. It is convenient to define $H_\mu^+ = \Theta(\mu - M\psi)$ and $H_\mu^- = \Theta(M\psi - \mu)$, where we use $\Theta(x)$ to denote the Heaviside Theta symbol, equal to 1 for $x > 0$ and zero otherwise. We can write

$$
\tilde{\mathcal{L}} \approx \mathcal{L} + \sum_{ij} \sum_{\mu\nu} \mathcal{Q}_{ij}^{\mu\nu} \frac{\epsilon_{i\mu} \epsilon_{j\nu}}{2}
\tag{E12}
$$

since the first-order terms vanish. The quadratic form writes:

$$
\begin{aligned}
\mathcal{Q}_{ij}^{\mu\nu} &= (U_i^+ U_j^+ + U_i^- U_j^-)(1 - \tanh^2(W_+) H_\mu^+ - \tanh^2(W_-) H_\mu^-) \delta_{\mu\nu} \\
&\quad - \left( 1 - \frac{1+x}{2} m_+^2 - \frac{1-x}{2} m_-^2 \right) (n_+^2 H_\mu^+ H_\nu^+ + n_-^2 H_\mu^- H_\nu^-) \delta_{ij} \\
&\quad - m_-^2 n_+^2 U_i^- U_j^- H_\mu^+ H_\nu^+ - m_+^2 n_-^2 U_i^+ U_j^+ H_\mu^- H_\nu^- \\
&\quad - \frac{1}{2} m_+ m_- n_+ n_- (U_i^- U_j^+ H_\mu^+ H_\nu^- + U_i^+ U_j^- H_\mu^- H_\nu^+) \\
&\quad - N\gamma \delta_{ij} \delta_{\mu\nu}
\end{aligned}
\tag{E13}
$$

To save some space, let's denote

$$
T_\pm = 1 - \tanh^2(W_\pm), \quad A = 1 - \frac{1+x}{2} m_+^2 - \frac{1-x}{2} m_-^2, \quad C = \frac{1}{2} m_+ m_- n_+ n_-, \quad B_\pm = m_\mp^2 n_\pm^2
\tag{E14}
$$

Then,

$$
\mathcal{Q}_{ij}^{\mu\nu} =
\begin{cases}
T_+ (U_i^+ U_j^+ + U_i^- U_j^-) \delta_{\mu\nu} - B_+ U_i^- U_j^- - A n_+^2 \delta_{ij} - N\gamma \delta_{ij} \delta_{\mu\nu} & \mu, \nu > M\psi \\
T_- (U_i^+ U_j^+ + U_i^- U_j^-) \delta_{\mu\nu} - B_- U_i^+ U_j^+ - A n_-^2 \delta_{ij} - N\gamma \delta_{ij} \delta_{\mu\nu} & \mu, \nu < M\psi \\
-C U_i^- U_j^+ & \mu > M\psi > \nu \\
-C U_i^+ U_j^- & \mu < M\psi < \nu
\end{cases}
\tag{E15}
$$

We would like to diagonalize this form, that is, find eigenvectors $\epsilon_{i\mu}$ with their corresponding eigenvalues $\lambda$, such that:

$$\sum_{j\nu} \mathcal{Q}_{ij}^{\mu\nu} \epsilon_{j\nu} = \lambda \epsilon_{i\mu} \tag{E16}$$

If we can show that all such eigenvalues are negative, we will have established that the rank-2 weights (E1) we have found are indeed a local maximum of the log-likelihood.

### 1. Symmetric eigenvectors

The form $\mathcal{Q}_{ij}^{\mu\nu}$ is invariant under permutation of hidden units within the $+$ and $-$ groups (that is, permutations that preserve the inequalities $\mu > M\psi$ and $\mu < M\psi$). We therefore propose eigenvectors of the form:

$$\epsilon_{i\mu} = \epsilon_i^+ H_\mu^+ + \epsilon_i^- H_\mu^- = \begin{cases} \epsilon_i^+ & \mu > M\psi \\ \epsilon_i^- & \mu < M\psi \end{cases} \tag{E17}$$

These vectors fully preserve the symmetry of hidden units within a group. Substituting this eigenvector proposal, we find

$$\sum_{j\nu} \mathcal{Q}_{ij}^{\mu\nu} \epsilon_{j\nu} = \sum_j \sum_{\nu > M\psi} \mathcal{Q}_{ij}^{\mu\nu} \epsilon_j^+ + \sum_j \sum_{\nu < M\psi} \mathcal{Q}_{ij}^{\mu\nu} \epsilon_j^-$$

$$= \begin{cases} T_+ \sum_j \epsilon_j^+ (U_i^+ U_j^+ + U_i^- U_j^-) - M_+ B_+ \sum_j \epsilon_j^+ U_i^- U_j^- - (M_+ An_+^2 + N\gamma)\epsilon_i^+ - M_- C \sum_j U_i^- U_j^+ \epsilon_j^- & \mu > M\psi \\ T_- \sum_j \epsilon_j^- (U_i^+ U_j^+ + U_i^- U_j^-) - M_- B_- \sum_j \epsilon_j^- U_i^+ U_j^+ - (M_- An_-^2 + N\gamma)\epsilon_i^- - M_+ C \sum_j U_i^+ U_j^- \epsilon_j^+ & \mu < M\psi \end{cases} \tag{E18}$$

To have an eigenvector here, this must be equal to $\lambda \epsilon_{i\mu}$, where $\lambda$ is an eigenvalue. We can then write the eigen-equations:

$$T_+ \sum_j \epsilon_j^+ (U_i^+ U_j^+ + U_i^- U_j^-) - M_+ B_+ \sum_j \epsilon_j^+ U_i^- U_j^- - M_- C \sum_j U_i^- U_j^+ \epsilon_j^- = (\lambda + M_+ An_+^2 + N\gamma)\epsilon_i^+$$

$$T_- \sum_j \epsilon_j^- (U_i^+ U_j^+ + U_i^- U_j^-) - M_- B_- \sum_j \epsilon_j^- U_i^+ U_j^+ - M_+ C \sum_j U_i^+ U_j^- \epsilon_j^+ = (\lambda + M_- An_-^2 + N\gamma)\epsilon_i^- \tag{E19}$$

The most general $\epsilon_i^\pm$ can be written in the form $\epsilon_i^\pm = \zeta_+^\pm U_i^+ + \zeta_-^\pm U_i^- + \zeta_\perp^\pm Y_i^\pm$, where $Y_i^\pm$ are some vectors perpendicular to the data, that is, $\sum_i Y_i^\pm V_i^k = 0$. The coefficients $\zeta$'s are to be determined. Recalling the squared norm of $U_i^\pm$ is $N\frac{1\pm x}{2}$, it follows that:

$$NT_+ \left( \frac{1+x}{2}\zeta_+^+ U_i^+ + \frac{1-x}{2}\zeta_-^+ U_i^- \right) - NM_+ B_+ \frac{1-x}{2}\zeta_-^+ U_i^- - NM_- C \frac{1+x}{2}\zeta_+^- U_i^-$$
$$= (\lambda + M_+ An_+^2 + N\gamma)(\zeta_+^+ U_i^+ + \zeta_-^+ U_i^- + \zeta_\perp^+ Y_i^+) \tag{E20}$$

and

$$NT_- \left( \frac{1+x}{2}\zeta_+^- U_i^+ + \frac{1-x}{2}\zeta_-^- U_i^- \right) - NM_- B_- \frac{1+x}{2}\zeta_+^- U_i^+ - NM_+ C \frac{1-x}{2}\zeta_-^+ U_i^+$$
$$= (\lambda + M_- An_-^2 + N\gamma)(\zeta_+^- U_i^+ + \zeta_-^- U_i^- + \zeta_\perp^- Y_i^-) \tag{E21}$$

Considering the linearly independent components, these equations break apart into:

$$\left( NT_+ \frac{1+x}{2} - \lambda - M_+ An_+^2 - N\gamma \right) \zeta_+^+ = 0, \quad \text{(along } U_i^+, \mu > M\psi)$$

$$\left( NT_- \frac{1-x}{2} - \lambda - M_- An_-^2 - N\gamma \right) \zeta_-^- = 0, \quad \text{(along } U_i^-, \mu < M\psi)$$

$$\left( N(T_+ - M_+ B_+) \frac{1-x}{2} - \lambda - M_+ An_+^2 - N\gamma \right) \zeta_-^+ - NM_- C \frac{1+x}{2}\zeta_+^- = 0, \quad \text{(along } U_i^-, \mu > M\psi) \tag{E22}$$

$$\left( N(T_- - M_- B_-) \frac{1+x}{2} - \lambda - M_- An_-^2 - N\gamma \right) \zeta_+^- - NM_+ C \frac{1-x}{2}\zeta_-^+ = 0, \quad \text{(along } U_i^+, \mu < M\psi)$$

$$(\lambda + M_+ An_+^2 + N\gamma)\zeta_\perp^+ = 0, \quad \text{(along } Y_i^+, \mu > M\psi)$$

$$(\lambda + M_- An_-^2 + N\gamma)\zeta_\perp^- = 0, \quad \text{(along } Y_i^-, \mu < M\psi)$$

Here we are careful to not eliminate the coefficients $\zeta$'s because they can be zero! Indeed, we have the following possibilities:

- $\zeta_+^+$ is non-zero. Then $\lambda_+^+ = NT_+\frac{1+x}{2} - M_+An_+^2 - N\gamma$. All the other coefficients $\zeta$'s vanish. The perturbation is $\epsilon_{i\mu} \propto U_i^+ H_\mu^+$.

- $\zeta_-^-$ is non-zero. Then $\lambda_-^- = NT_-\frac{1-x}{2} - M_-An_-^2 - N\gamma$. All the other coefficients $\zeta$'s vanish. The perturbation is $\epsilon_{i\mu} \propto U_i^- H_\mu^-$.

- $\zeta_\perp^+$ is non-zero. Then $\lambda_\perp^+ = -M_+An_+^2 - N\gamma$. All the other coefficients $\zeta$'s vanish. This has degeneracy $N-2$ for all the directions that $Y_i^+$ can take. The perturbation is $\epsilon_{i\mu} \propto Y_i^+ H_\mu^+$.

- $\zeta_\perp^-$ is non-zero. Then $\lambda_\perp^- = -M_-An_-^2 - N\gamma$. All the other coefficients $\zeta$'s vanish. This has degeneracy $N-2$ for all the directions that $Y_i^-$ can take. The perturbation is $\epsilon_{i\mu} \propto Y_i^- H_\mu^-$.

The only remaining possibility has $\zeta_-^+, \zeta_+^-$ non-zero, and all the other coefficients $\zeta_+^+ = \zeta_-^- = \zeta_\perp^+ = \zeta_\perp^- = 0$ zero. The perturbation is of the "mix" form $\epsilon_{i\mu} = \zeta_+^- U_i^+ H_\mu^- + \zeta_-^+ U_i^- H_\mu^+$. These perturbations can be interpreted as a partial displacement of hidden units from being aligned to $U_i^+$, towards being aligned to $U_i^-$ (or vice-versa). In this case we obtain a $2 \times 2$ system of equations:

$$
\begin{aligned}
\left(N(T_+ - M_+B_+)\frac{1-x}{2} - M_+An_+^2 - N\gamma\right)\zeta_-^+ - NM_-C\frac{1+x}{2}\zeta_+^- = \lambda\zeta_-^+ \\
\left(N(T_- - M_-B_-)\frac{1+x}{2} - M_-An_-^2 - N\gamma\right)\zeta_+^- - NM_+C\frac{1-x}{2}\zeta_-^+ = \lambda\zeta_+^-
\end{aligned}
\tag{E23}
$$

Defining $a_\pm = N(T_\pm - M_\pm B_\pm)\frac{1\mp x}{2} - M_\pm An_\pm^2 - N\gamma$ and $b_\pm = NM_\mp C\frac{1\pm x}{2}$, we can write this as:

$$
\begin{aligned}
a_+\zeta_-^+ - b_+\zeta_+^- = \lambda\zeta_-^+ \\
a_-\zeta_+^- - b_-\zeta_-^+ = \lambda\zeta_+^-
\end{aligned}
\tag{E24}
$$

This has the two eigenvalues (we won't write the corresponding eigenvectors as they are messy),

$$
\lambda_\pm^{\text{mix}} = \frac{1}{2}\left(a_+ + a_- \pm \sqrt{(a_+ - a_-)^2 + 4b_+b_-}\right)
\tag{E25}
$$

Substituting,

$$
\begin{aligned}
\lambda_\pm^{\text{mix}} = \frac{N}{2}\left((T_+ - M_+B_+)\frac{1-x}{2} + (T_- - M_-B_-)\frac{1+x}{2} - A(\alpha_+n_+^2 + \alpha_-n_-^2) - 2\gamma\right) \\
\pm \frac{N}{2}\sqrt{\left((T_+ - M_+B_+)\frac{1-x}{2} - (T_- - M_-B_-)\frac{1+x}{2} - A(\alpha_+n_+^2 - \alpha_-n_-^2)\right)^2 + 4M_+M_-C^2\frac{1+x}{2}\frac{1-x}{2}}
\end{aligned}
\tag{E26}
$$

Recalling the definitions

$$
T_\pm = 1 - \tanh^2(W_\pm), \quad A = 1 - \frac{1+x}{2}m_+^2 - \frac{1-x}{2}m_-^2, \quad C = \frac{1}{2}m_+m_-n_+n_-, \quad B_\pm = m_\mp^2 n_\pm^2
\tag{E27}
$$

We then have the large $N$ dominant behavior:

$$
\begin{aligned}
\lambda_\pm^{\text{mix}} = -\frac{N}{2}\left(M_+B_+\frac{1-x}{2} + M_-B_-\frac{1+x}{2}\right) \\
\pm \frac{N}{2}\sqrt{\left(M_+B_+\frac{1-x}{2} - M_-B_-\frac{1+x}{2}\right)^2 + M_+M_-B_+B_-\frac{1+x}{2}\frac{1-x}{2}}
\end{aligned}
\tag{E28}
$$

where we note that $4C^2 = B_+B_-$.

## 2. Eigenvectors that break hidden unit symmetry

In the previous section we have found $2N$ eigenvectors. We must find in total $NM$, thus to find the remaining $N(M-2)$, we must break the symmetry of the hidden units in each group. We may consider perturbations of the form:

$$\epsilon_{i\mu} = \epsilon_i^+ W_\mu^+ H_\mu^+ + \epsilon_i^- W_\mu^- H_\mu^- = \begin{cases} \epsilon_i^+ W_\mu^+ & \mu > M\psi \\ \epsilon_i^- W_\mu^- & \mu < M\psi \end{cases} \tag{E29}$$

with some $W_\mu^\pm$ that carry the newly introduced differentiation of the hidden units. These perturbations should be orthogonal to those already considered, so we must require $\sum_\mu H_\mu^\pm \epsilon_{i\mu} = 0$, which implies that

$$\sum_{\mu > M\psi} W_\mu^+ = \sum_{\mu < M\psi} W_\mu^- = 0 \tag{E30}$$

Now recall the full expression of the tensor $\mathcal{Q}_{ij}^{\mu\nu}$,

$$\mathcal{Q}_{ij}^{\mu\nu} = \begin{cases} T_+(U_i^+ U_j^+ + U_i^- U_j^-)\delta_{\mu\nu} - B_+ U_i^- U_j^- - An_+^2 \delta_{ij} - N\gamma\delta_{ij}\delta_{\mu\nu} & \mu,\nu > M\psi \\ T_-(U_i^+ U_j^+ + U_i^- U_j^-)\delta_{\mu\nu} - B_- U_i^+ U_j^+ - An_-^2 \delta_{ij} - N\gamma\delta_{ij}\delta_{\mu\nu} & \mu,\nu < M\psi \\ -C U_i^- U_j^+ & \mu > M\psi > \nu \\ -C U_i^+ U_j^- & \mu < M\psi < \nu \end{cases} \tag{E31}$$

When forming the product $\sum_{j\nu} \mathcal{Q}_{ij}^{\mu\nu} \epsilon_{j\nu}$, any sum of $\epsilon_{j\nu}$ over $\nu > M\psi$ or $\nu < M\psi$ will vanish. This observation implies that

$$\sum_{j\nu} \mathcal{Q}_{ij}^{\mu\nu} \epsilon_{j\nu} = \sum_{j\nu} \tilde{\mathcal{Q}}_{ij}^{\mu\nu} \epsilon_{j\nu} \tag{E32}$$

where we have removed from the "effective" tensor $\tilde{\mathcal{Q}}$ any contributions that are proportional to $H_\nu^\pm$,

$$\tilde{\mathcal{Q}}_{ij}^{\mu\nu} = \begin{cases} T_+(U_i^+ U_j^+ + U_i^- U_j^-)\delta_{\mu\nu} - N\gamma\delta_{ij}\delta_{\mu\nu} & \mu,\nu > M\psi \\ T_-(U_i^+ U_j^+ + U_i^- U_j^-)\delta_{\mu\nu} - N\gamma\delta_{ij}\delta_{\mu\nu} & \mu,\nu < M\psi \\ 0 & \text{otherwise} \end{cases} \tag{E33}$$

With this in mind, it follows that $\sum_{j\nu} \mathcal{Q}_{ij}^{\mu\nu} \epsilon_{j\nu}$ simplifies to:

$$\begin{aligned} \sum_{j\nu} \mathcal{Q}_{ij}^{\mu\nu} \epsilon_{j\nu} &= \sum_j [T_+(U_i^+ U_j^+ + U_i^- U_j^-) - N\gamma\delta_{ij}]\epsilon_j^+ W_\mu^+ H_\mu^+ \\ &+ \sum_j [T_-(U_i^+ U_j^+ + U_i^- U_j^-) - N\gamma\delta_{ij}]\epsilon_j^- W_\mu^- H_\mu^- \end{aligned} \tag{E34}$$

Recalling that $\sum_i (U_i^\pm)^2 = N\frac{1\pm x}{2}$, and that $W_\mu^+, W_\mu^-$ each span $M_+ - 1$ and $M_- - 1$ dimensional spaces, respectively, we can write down immediately the 6 eigenvectors:

- $\epsilon_i^+ \propto U_i^+$ and $\epsilon_i^- = 0$, with eigenvalue $\lambda_{++}^b = NT_+\frac{1+x}{2} - N\gamma$. The $W_\mu^+$ are free to span a $M_+ - 1$-dimensional space subject to $\sum_{\mu > M\psi} W_\mu^+ = 0$.

- $\epsilon_i^+ \propto U_i^-$ and $\epsilon_i^- = 0$, with eigenvalue $\lambda_{-+}^b = NT_+\frac{1-x}{2} - N\gamma$. The $W_\mu^+$ are free to span a $M_+ - 1$-dimensional space.

- $\epsilon_i^- \propto U_i^+$ and $\epsilon_i^+ = 0$, with eigenvalue $\lambda_{+-}^b = NT_-\frac{1+x}{2} - N\gamma$. The $W_\mu^-$ are free to span a $M_- - 1$-dimensional space.

- $\epsilon_i^- \propto U_i^-$ and $\epsilon_i^+ = 0$, with eigenvalue $\lambda_{--}^b = NT_-\frac{1-x}{2} - N\gamma$. The $W_\mu^-$ are free to span a $M_- - 1$-dimensional space.

- $\epsilon_i^+, \epsilon_i^-$ are both perpendicular to both $U_i^+$ and $U_i^-$, spanning a $N - 2$-dimensional space. The eigenvalue equals $-N\gamma$. The $W_\mu^+, W_\mu^-$ together span a $M - 2$-dimensional space.

Summing the multiplicities of these eigenvalues, we have a total of $2M - 4 + (N-2)(M-2) = N(M-2)$, indeed just what we were missing.

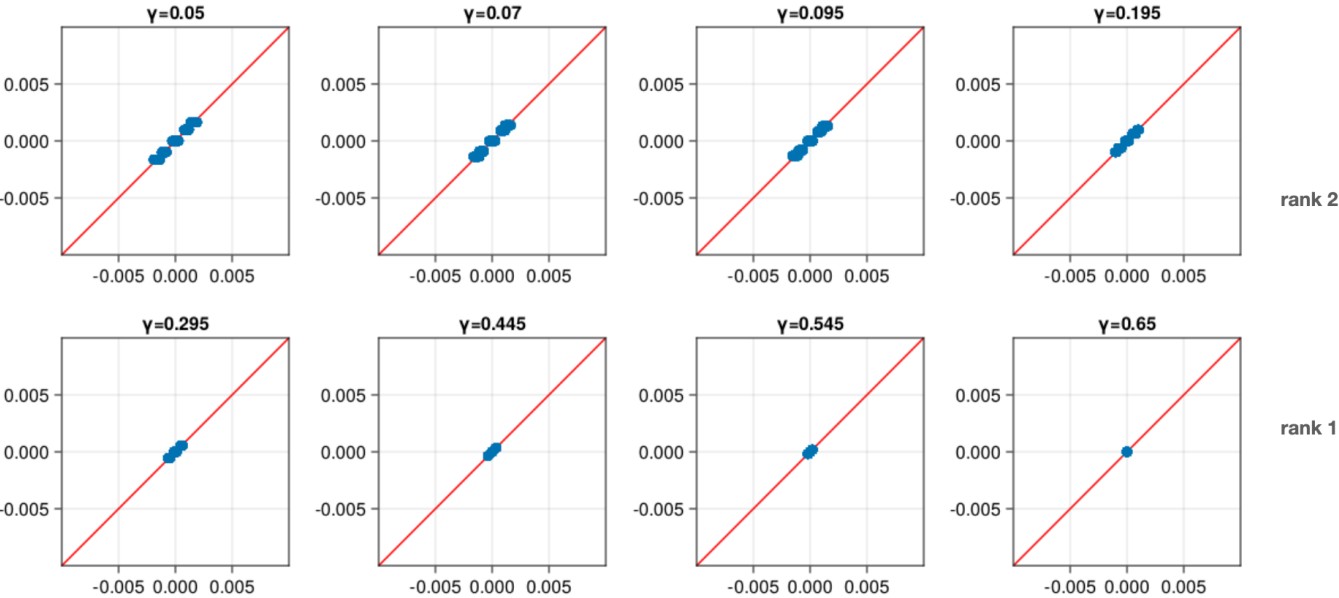

FIG. S2. Comparison of Rank-2 *ansatz* to weights obtained from training RBMs on two data points with different values of the regularization $\gamma$.

### 3. Signs of the eigenvalues

Now we want to check whether these eigenvectors we have found have negative eigenvalues. The perpendicular modes $\lambda_\perp^\pm < 0$ are manifestly negative. Similarly, $\lambda_-^{\mathrm{mix}} < 0$ is clearly negative. For $\lambda_+^{\mathrm{mix}}$, noting that $4C^2 = B_+ B_-$, we can write:

$$
\begin{aligned}
\lambda_+^{\mathrm{mix}} &= -\frac{N}{2}\left(M_+ B_+ \frac{1-x}{2} + M_- B_- \frac{1+x}{2}\right) \\
&\quad + \frac{N}{2}\sqrt{\left(M_+ B_+ \frac{1-x}{2} + M_- B_- \frac{1+x}{2}\right)^2 - 3 M_+ M_- B_+ B_- \frac{1+x}{2}\frac{1-x}{2}} \\
&\leq -\frac{N}{2}\left(M_+ B_+ \frac{1-x}{2} + M_- B_- \frac{1+x}{2}\right) + \frac{N}{2}\sqrt{\left(M_+ B_+ \frac{1-x}{2} + M_- B_- \frac{1+x}{2}\right)^2} \\
&= 0
\end{aligned}
\tag{E35}
$$

and thus is also negative.

The modes $\lambda_+^+, \lambda_-^-$ correspond to perturbations that do not change the form of the rank-2 solution we have proposed. Since we claim this solution to be a maximum among weights of this form, at least within solutions of this form, these modes should be negative!

Finally consider the modes breaking the hidden unit symmetry. We observe that $\lambda_+^+ = \lambda_{++}^{\mathrm{b}} - M_+ A n_+^2 \leq \lambda_{++}^{\mathrm{b}}$ and $\lambda_-^- = \lambda_{--}^{\mathrm{b}} - M_- A n_-^2 \leq \lambda_{--}^{\mathrm{b}}$. Also, since we assume that $x \geq 0$, we have that $\lambda_{-+}^{\mathrm{b}} \leq \lambda_{++}^{\mathrm{b}}$ and $\lambda_{--}^{\mathrm{b}} \leq \lambda_{+-}^{\mathrm{b}}$. Therefore, to establish the stability of a solution, it suffices to consider the eigenvalues $\lambda_{++}^{\mathrm{b}}$ and $\lambda_{+-}^{\mathrm{b}}$. We have checked numerically that these pair of eigenvalues are negative, across a wide range of values of $x, \alpha, \gamma$.

### 4. Numerical verification

To conclude, we have shown that the weights (E1) give a stable local maxima of the regularized log-likelihood. We cannot rule out the possibility of a better global maximum, but this seems unlikely. We also report numerical simulations of RBMs trained by persistent contrastive divergence at various levels of regularization and which show excellent agreement with (E1). See Figure S2.

## 5. Phase transition from rank-1 to rank-2 weights

For large enough $\gamma$ the weights are are of rank-1, thus $\psi = 0$ in (E1). In this case all hidden units align along the principal direction $U_i^+$ (recall we assume throughout $x > 0$). By repeating the above perturbative analysis around the rank-1 weights, we find that this solution eventually becomes unstable to perturbations $U_i^-$ that affect some of the hidden units and thus break their equivalence.

This is a genuine phase transition, where $\psi$ passes from being zero, to taking non-zero values as the weights become of rank 2. To find the critical parameters separating the two regimes, we can differentiate the log-likelihood with respect to $\psi$, and evaluate at $\psi = 0$. This gradient is negative in the rank-1 regime (where increasing $\psi$ is detrimental to the likelihood), but becomes positive in the rank-2 phase. Thus setting $\partial LL/\partial \psi = 0$, gives the equation:

$$
\begin{aligned}
\ln \cosh(W_+) - \ln \cosh(W_+ m_+) - \frac{W_+^2 \gamma}{1+x} &= \\
\ln \cosh(W_-) - \ln \cosh(W_- m_-) - \frac{W_-^2 \gamma}{1-x}
\end{aligned}
\tag{E36}
$$

where $W_\pm, m_\pm$ are the solutions of the previous saddle-point equations. These equations determine critical values of $x, \gamma, \alpha$ at the boundary between the rank-1 and rank-2 phases.

## Appendix F: Computation of $\tau_{\text{swap}}$

As in the main text, we consider two stacked RBMs:

- Bottom RBM, with $N^0 = N$ visible units and $N^1 = \alpha^1 N$ hidden units.

- Top RBM, with $N^1 = \alpha^1 N$ visible units and $N^2 = \alpha^1 \alpha^2 N$ hidden units.

Let $(\mathbf{v}^1, \mathbf{h}^1)$ and $(\mathbf{v}^2, \mathbf{h}^2)$ be visible and hidden configurations of the bottom and top RBMs, respectively. The probability of accepting a swap $\mathbf{h}^1 \leftrightarrow \mathbf{v}^2$ is given by the Metropolis rule,

$$
A(\mathbf{h}^1, \mathbf{v}^2) = \min \left( 1, \frac{P_h^1(\mathbf{v}^2) P_v^2(\mathbf{h}^1)}{P_h^1(\mathbf{h}^1) P_v^2(\mathbf{v}^2)} \right)
\tag{F1}
$$

where $P_v^\ell(\mathbf{z})$ and $P_h^\ell(\mathbf{z})$ are the marginal probabilities of the visible and hidden configurations of the two RBMs, $\ell = 1, 2$. We want to compute the average acceptance probability,

$$
\begin{aligned}
\langle A \rangle &= \sum_{\mathbf{h}^1, \mathbf{v}^2} P_h^1(\mathbf{h}^1) P_v^2(\mathbf{v}^2) A(\mathbf{h}^1, \mathbf{v}^2) \\
&= \sum_{\mathbf{h}^1, \mathbf{v}^2} \min \left( P_h^1(\mathbf{h}^1) P_v^2(\mathbf{v}^2), P_h^1(\mathbf{v}^2) P_v^2(\mathbf{h}^1) \right)
\end{aligned}
\tag{F2}
$$

The typical swap time, $\tau_{\text{swap}}$, can be estimated as the inverse of $\langle A \rangle$.

We assume here that the weights $W_{ij}^1, W_{jk}^2$ of the two RBMs are of the forms (E1), so

$$
W_{ij}^1 = \begin{cases} \frac{W_+^1}{N^0} \frac{V_i^1 + V_i^2}{1+x^0} & j > N^1 \psi^1 \\ \frac{W_-^1}{N^0} \frac{V_i^1 - V_i^2}{1-x^0} & j < N^1 \psi^1 \end{cases}, \quad
W_{jk}^2 = \begin{cases} \frac{W_+^2}{N^1} \frac{H_j^1 + H_j^2}{1+x^1} & k > N^2 \psi^2 \\ \frac{W_-^2}{N^1} \frac{H_j^1 - H_j^2}{1-x^1} & k < N^2 \psi^2 \end{cases}
\tag{F3}
$$

Here $V_i^1, V_i^2$ refer to the two data points on which the bottom RBM was trained, while $H_j^1, H_j^2$ refer to their most probable "images", that is, the most probable states of the hidden units of the bottom RBM, when conditioned on one data point, or the other. These are given by:

$$
H_j^1 = 1, \quad H_j^2 = \begin{cases} +1 & j > N^1 \psi^1 \\ -1 & j < N^1 \psi^1 \end{cases}
\tag{F4}
$$

That is, we assume that the top RBM has been trained on the modes of the representations of the two bottom data points (so we ignore fluctuations in these representations). We take care not to confuse the upper indices in $V_i^1, V_i^2$

and $H_j^1, H_j^2$, which refer to the two data points, with the upper indices used elsewhere to refer to the two RBMs of the stack.

In turn, the most likely hidden unit configurations of the top RBM, when conditioned on $H_j^1, H_j^2$ at its visible layer, are given by

$$G_k^1 = 1, \quad G_k^2 = \begin{cases} +1 & k > N^2\psi^2 \\ -1 & k < N^2\psi^2 \end{cases} \tag{F5}$$

We also define the overlaps between the data points and their representations at each level of the stack:

$$x^0 = \frac{1}{N^0} \sum_i V_i^1 V_i^2 = 1 - 2\psi^0$$

$$x^1 = \frac{1}{N^1} \sum_j H_j^1 H_j^2 = 1 - 2\psi^1 \tag{F6}$$

$$x^2 = \frac{1}{N^2} \sum_k G_k^1 G_k^2 = 1 - 2\psi^2$$

We can assume without losing generality that $V_i^1 = 1$ for all $i$, and that $V_i^2 = 1$ or $V_i^2 = -1$ according to whether $i > N^0\psi^0$ or $i < N^0\psi^0$, with $x^0 = 1 - 2\psi^0$ (see below). This can always be arranged by re-defining the signs of some sites $i$, and by re-ordering indexes $i$ appropriately. Then the weights can be writen in block-form:

$$W_{ij}^1 = \begin{cases} \frac{W_+^1}{N_+^0} & i > N^0\psi^0, j > N^1\psi^1 \\ \frac{W_-^1}{N_-^0} & i < N^0\psi^0, j < N^1\psi^1 \\ 0 & \text{otherwise} \end{cases}, \quad W_{jk}^2 = \begin{cases} \frac{W_+^2}{N_+^1} & j > N^1\psi^1, k > N^2\psi^2 \\ \frac{W_-^2}{N_-^1} & j < N^1\psi^1, k < N^2\psi^2 \\ 0 & \text{otherwise} \end{cases} \tag{F7}$$

where $N_\pm^\ell = N^\ell \frac{1 \pm x^\ell}{2}$. That is, the weight matrices $W^1$ and $W^2$ are block-diagonal. On the bottom RBM, the visible units $i > N^0\psi^0$ are only connected to hidden units $j > N^1\psi^1$, while the visible units $i < N^0\psi^0$ are only connected to hidden units $j < N^1\psi^1$. The RBM decouples into two independent "sub-RBMs". Similar remarks hold of course for the top RBM.

The hidden layer of the bottom RBM and the visible layer of the top RBM have the same dimensions and can exchange their configurations. For a configuration $\mathbf{h}^1, \mathbf{v}^2$ of either of these layers, we can define the overlaps:

$$n_+(\mathbf{h}) = \frac{1}{N_+^1} \sum_{j > N^1\psi^1} h_j, \quad n_-(\mathbf{h}) = \frac{1}{N_-^1} \sum_{j < N^1\psi^1} h_j \tag{F8}$$

With these definitions, we have that:

$$P_h^1(\mathbf{h}^1) = \frac{e^{-N^1 \mathcal{E}_h^1(\mathbf{h}^1)}}{Z_1}, \quad P_v^2(\mathbf{v}^2) = \frac{e^{-N^1 \mathcal{E}_v^2(\mathbf{v}^2)}}{Z_2} \tag{F9}$$

where $Z_1, Z_2$ are the partition functions of the two RBMs, and $\mathcal{E}_h^1(\mathbf{h}^1), \mathcal{E}_v^2(\mathbf{v}^2)$ are the effective energies in each layer, divided by the number of units in the layer:

$$\begin{aligned} -\mathcal{E}_h^1(\mathbf{h}^1) &= \frac{1}{N^1} \sum_i \ln 2 \cosh\left(\sum_j W_{ij}^1 h_j^1\right) \\ &= \frac{\psi_+^0}{\alpha^1} \ln \cosh\left(\frac{W_+^1}{N_+^0} \sum_{j > N^1\psi^1} h_j^1\right) + \frac{\psi_-^0}{\alpha^1} \ln \cosh\left(\frac{W_-^1}{N_-^0} \sum_{j < N^1\psi^1} h_j^1\right) + \frac{\ln 2}{\alpha^1} \\ &= \frac{\psi_+^0}{\alpha^1} \ln \cosh\left(\frac{\psi_+^1}{\psi_+^0} \alpha^1 W_+^1 n_+(\mathbf{h}^1)\right) + \frac{\psi_-^0}{\alpha^1} \ln \cosh\left(\frac{\psi_-^1}{\psi_-^0} \alpha^1 W_-^1 n_-(\mathbf{h}^1)\right) + \frac{\ln 2}{\alpha^1} \end{aligned} \tag{F10}$$

and

$$
\begin{aligned}
-\mathcal{E}_v^2(\mathbf{v}^2) &= \frac{1}{N^1} \sum_k \ln 2 \cosh \left( \sum_j W_{jk}^2 v_j^2 \right) \\
&= \alpha^2 \psi_+^2 \ln \cosh \left( \frac{W_+^2}{N_+^1} \sum_{j > N^1 \psi^1} v_j^2 \right) + \alpha^2 \psi_-^2 \ln \cosh \left( \frac{W_-^2}{N_-^1} \sum_{j < N^1 \psi^1} v_j^2 \right) + \alpha^2 \ln 2 \\
&= \alpha^2 \psi_+^2 \ln \cosh \left( W_+^2 n_+(\mathbf{v}^2) \right) + \alpha^2 \psi_-^2 \ln \cosh \left( W_-^2 n_-(\mathbf{v}^2) \right) + \alpha^2 \ln 2
\end{aligned}
\tag{F11}
$$

where $\psi_+^\ell = 1 - \psi^\ell$ and $\psi_-^\ell = \psi^\ell$. Notice that the dependence on the configurations $\mathbf{h}^1, \mathbf{v}^2$ occurs only through the overlaps $n_\pm(\mathbf{h}^1), n_\pm(\mathbf{v}^2)$. Let's denote $\mathbf{n} = (n_+(\mathbf{h}^1), n_-(\mathbf{h}^1))$ and $\mathbf{m} = (n_+(\mathbf{v}^2), n_-(\mathbf{v}^2))$, to distinguish them. We can define:

$$
\begin{aligned}
\mathcal{E}_h^1(\mathbf{n}) &= \frac{\psi_+^0}{\alpha^1} \mathcal{E}_+^1(n_+) + \frac{\psi_-^0}{\alpha^1} \mathcal{E}_-^1(n_-) \\
\mathcal{E}_v^2(\mathbf{m}) &= \alpha^2 \psi_+^2 \mathcal{E}_+^2(m_+) + \alpha^2 \psi_-^2 \mathcal{E}_-^2(m_-)
\end{aligned}
\tag{F12}
$$

with,

$$
\begin{aligned}
-\mathcal{E}_\pm^1(n_\pm) &= \ln \cosh \left( \frac{\psi_\pm^1}{\psi_\pm^0} \alpha^1 W_\pm^1 n_\pm \right) + \ln 2 \\
-\mathcal{E}_\pm^2(m_\pm) &= \ln \cosh(W_\pm^2 m_\pm) + \ln 2
\end{aligned}
\tag{F13}
$$

It follows that,

$$
\begin{aligned}
\langle A \rangle &= \sum_{\mathbf{h}^1, \mathbf{v}^2} \min \left( \frac{e^{-N^1 \mathcal{E}_h^1(\mathbf{h}^1)}}{Z_1} \frac{e^{-N^1 \mathcal{E}_v^2(\mathbf{v}^2)}}{Z_2}, \frac{e^{-N^1 \mathcal{E}_h^1(\mathbf{v}^2)}}{Z_1} \frac{e^{-N^1 \mathcal{E}_v^2(\mathbf{h}^1)}}{Z_2} \right) \\
&= \int \min \left( \frac{e^{-N^1 \mathcal{E}_h^1(\mathbf{n})}}{Z_1} \frac{e^{-N^1 \mathcal{E}_v^2(\mathbf{m})}}{Z_2}, \frac{e^{-N^1 \mathcal{E}_h^1(\mathbf{m})}}{Z_1} \frac{e^{-N^1 \mathcal{E}_v^2(\mathbf{n})}}{Z_2} \right) e^{N^1 [\mathcal{S}(\mathbf{n}) + \mathcal{S}(\mathbf{m})]} d\mathbf{n} d\mathbf{m}
\end{aligned}
\tag{F14}
$$

where we have defined the entropy:

$$
e^{N^1 \mathcal{S}(\mathbf{n})} = \sum_{\mathbf{h}} \delta \left( n_+ - \frac{1}{N_+^1} \sum_{j > N^1 \psi^1} h_j \right) \delta \left( n_- - \frac{1}{N_-^1} \sum_{j < N^1 \psi^1} h_j \right)
\tag{F15}
$$

with the sum over $\mathbf{h}$ going through all $2^{N_1}$ configurations of the contact layer. We note here the expression for this entropy:

$$
\mathcal{S}(\mathbf{n}) = \psi_+^1 \mathcal{S}_h^+(n_+) + \psi_-^1 \mathcal{S}_h^-(n_-)
\tag{F16}
$$

where

$$
\mathcal{S}_\pm(n_\pm) = \min_{\hat{n}} \left[ -n_\pm \hat{n}_\pm + \ln \cosh(\hat{n}_\pm) \right] + \ln 2
\tag{F17}
$$

Since the constraint acting on the units $j < N^1 \psi^1$ is independent from the constraint acting on the units $j > N^1 \psi^1$, the entropy decouples, with two independent additive contributions. Clearly $n_\pm = \tanh(\hat{n}_\pm)$, so that

$$
\begin{aligned}
\mathcal{S}_\pm(n_\pm) &= -n_\pm \operatorname{atanh}(n_\pm) + \ln \cosh \operatorname{atanh}(n_\pm) + \ln 2 \\
&= -\frac{1 + n_\pm}{2} \ln \left( \frac{1 + n_\pm}{2} \right) - \frac{1 - n_\pm}{2} \ln \left( \frac{1 - n_\pm}{2} \right)
\end{aligned}
\tag{F18}
$$

equals the Spin entropy. Defining the action

$$
\mathcal{A}(\mathbf{n}, \mathbf{m}) = \max \{ \mathcal{E}_h^1(\mathbf{n}) + \mathcal{E}_v^2(\mathbf{m}), \mathcal{E}_h^1(\mathbf{m}) + \mathcal{E}_v^2(\mathbf{n}) \} - \mathcal{S}(\mathbf{n}) - \mathcal{S}(\mathbf{m})
\tag{F19}
$$

it follows that

$$\langle A \rangle = \frac{1}{Z_1 Z_2} \int e^{-N^1 \mathcal{A}(\mathbf{n}, \mathbf{m})} d\mathbf{n} d\mathbf{m} \tag{F20}$$

For large $N$ this is dominated by the $\mathbf{n}, \mathbf{m}$ that minimizes $\mathcal{A}$. Notice that the total entropy $\mathcal{S}(\mathbf{n}) + \mathcal{S}(\mathbf{m})$ remains the same before and after the swap. At the minimum of $\mathcal{A}$, the energies before and after the swap must coincide:

$$\mathcal{E}_h^1(\mathbf{n}) + \mathcal{E}_v^2(\mathbf{m}) = \mathcal{E}_h^1(\mathbf{m}) + \mathcal{E}_v^2(\mathbf{n}) \tag{F21}$$

i.e., at such a point there is no energy cost to swapping the configurations. Clearly $\mathbf{n} = \mathbf{m}$ satisfies this condition. Let's minimize $\mathcal{A}$, subject to this condition. We introduce a Lagrangian, of the form

$$\mathcal{L}(\mathbf{n}, \mathbf{m}; \lambda) = \mathcal{E}_h^1(\mathbf{n}) + \mathcal{E}_v^2(\mathbf{m}) - \mathcal{S}(\mathbf{n}) - \mathcal{S}(\mathbf{m}) - \lambda \left[ \mathcal{E}_h^1(\mathbf{n}) + \mathcal{E}_v^2(\mathbf{m}) - \mathcal{E}_h^1(\mathbf{m}) - \mathcal{E}_v^2(\mathbf{n}) \right] \tag{F22}$$

that we minimize over $\mathbf{n}, \mathbf{m}$, while setting the Lagrange multiplier $\lambda$ so as to satisfy the previous energy equality at the exchange. Rearranging terms, we can write

$$\mathcal{L} = \mathcal{L}(\mathbf{n}; \lambda) + \mathcal{L}(\mathbf{m}; 1 - \lambda) \tag{F23}$$

with

$$\mathcal{L}(\mathbf{n}; \lambda) = (1 - \lambda) \mathcal{E}_h^1(\mathbf{n}) + \lambda \mathcal{E}_v^2(\mathbf{n}) - \mathcal{S}(\mathbf{n})$$

Recall that

$$
\begin{aligned}
\mathcal{E}_h^1(\mathbf{n}) &= \frac{\psi_+^0}{\alpha^1} \mathcal{E}_+^1(n_+) + \frac{\psi_-^0}{\alpha^1} \mathcal{E}_-^1(n_-) \\
\mathcal{E}_v^2(\mathbf{m}) &= \alpha^2 \psi_+^2 \mathcal{E}_+^2(m_+) + \alpha^2 \psi_-^2 \mathcal{E}_-^2(m_-) \\
\mathcal{S}(\mathbf{n}) &= \psi_+^1 \mathcal{S}_+(n_+) + \psi_-^1 \mathcal{S}_-(n_-)
\end{aligned}
\tag{F24}
$$

with

$$
\begin{aligned}
\mathcal{E}_\pm^1(n_\pm) &= -\ln \cosh\left( \frac{\psi_\pm^1}{\psi_\pm^0} \alpha^1 W_\pm^1 n_\pm \right) - \ln 2 \\
\mathcal{E}_\pm^2(m_\pm) &= -\ln \cosh(W_\pm^2 m_\pm) - \ln 2 \\
\mathcal{S}_\pm(n_\pm) &= \min_{\hat{n}} \left[ -n_\pm \hat{n}_\pm + \ln \cosh(\hat{n}_\pm) \right] + \ln 2
\end{aligned}
\tag{F25}
$$

It follows that:

$$\mathcal{L}(\mathbf{n}; \lambda) = \mathcal{L}^+(n_+; \lambda) + \mathcal{L}^-(n_-; \lambda)$$

with

$$\mathcal{L}^\pm(n_\pm; \lambda) = (1 - \lambda) \frac{\psi_\pm^0}{\alpha^1} \mathcal{E}_\pm^1(n_\pm) + \lambda \alpha^2 \psi_\pm^2 \mathcal{E}_\pm^2(n_\pm) - \psi_\pm^1 \mathcal{S}_\pm(n_\pm) \tag{F26}$$

Therefore

$$\mathcal{L} = \mathcal{L}^+(n_+; \lambda) + \mathcal{L}^-(n_-; \lambda) + \mathcal{L}^+(m_+; 1 - \lambda) + \mathcal{L}^-(m_-; 1 - \lambda) \tag{F27}$$

Differentiating,

$$
\begin{aligned}
\frac{\partial \mathcal{L}}{\partial n_\pm} &= -(1 - \lambda) \psi_\pm^1 W_\pm^1 \tanh\left( \frac{\psi_\pm^1}{\psi_\pm^0} \alpha^1 W_\pm^1 n_\pm \right) - \lambda \alpha^2 \psi_\pm^2 W_\pm^2 \tanh(W_\pm^2 n_\pm) + \psi_\pm^1 \hat{n}_\pm \\
\frac{\partial \mathcal{L}}{\partial m_\pm} &= -\lambda \psi_\pm^1 W_\pm^1 \tanh\left( \frac{\psi_\pm^1}{\psi_\pm^0} \alpha^1 W_\pm^1 m_\pm \right) - (1 - \lambda) \alpha^2 \psi_\pm^2 W_\pm^2 \tanh(W_\pm^2 m_\pm) + \psi_\pm^1 \hat{m}_\pm
\end{aligned}
\tag{F28}
$$

We set these to zero and obtain saddle-point equations giving $\hat{n}_\pm, \hat{m}_\pm$ as function of the others. Then we also use $n_\pm = \tanh(\hat{n}_\pm)$ and $m_\pm = \tanh(\hat{m}_\pm)$ to complete the system of equations.

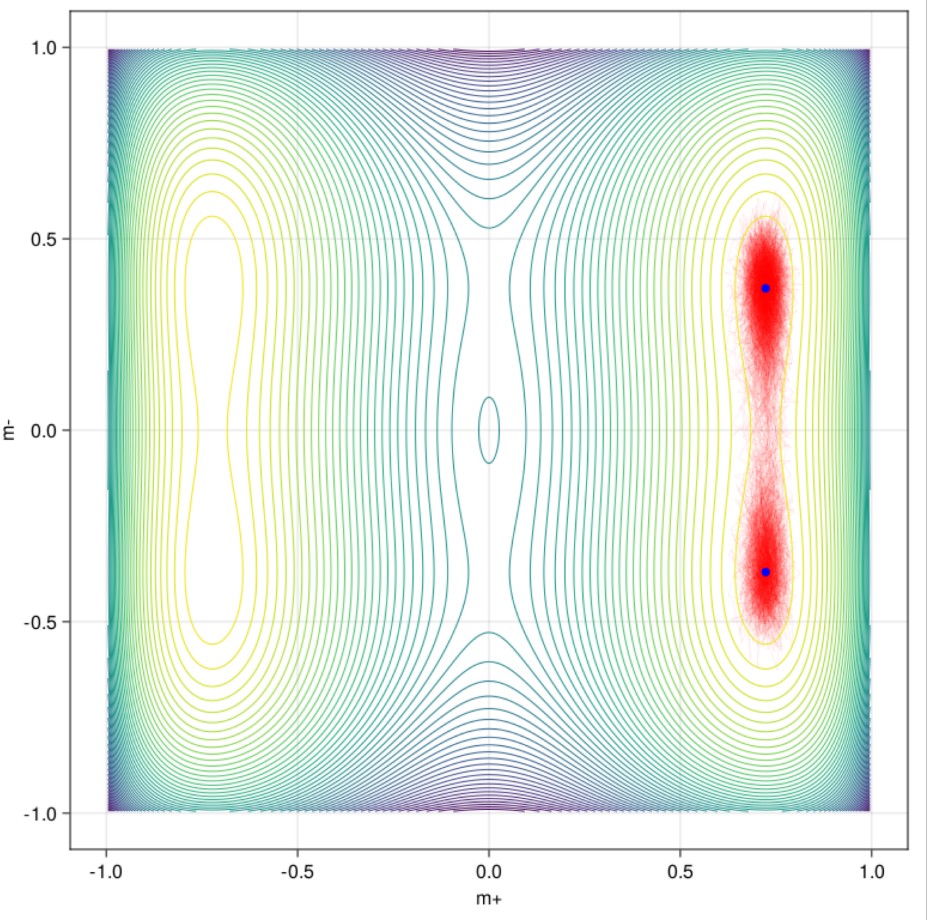

FIG. S3. Crossing between minima in a simple free-energy landscape.

To finish the calculation we need the partition functions $Z_1, Z_2$. But these follow for free, by setting $\lambda = 0$ in the previous Lagrangian.

Having determined these saddle-points, at $\lambda = \lambda^*$ (which satisfies the equivalent exchange energy condition) and at $\lambda = 0$ (for the partition functions), we have that, in the large $N$ limit,

$$\langle A \rangle \sim \frac{e^{-N^1 \mathcal{A}(\lambda^*)}}{e^{-N^1 \mathcal{A}(0)}}$$

where $\mathcal{A}(\lambda)$ denotes the saddle-point value,

$$\mathcal{A}(\lambda) = \mathcal{A}(\mathbf{n}, \mathbf{m}) = \mathcal{E}_h^1(\mathbf{n}) + \mathcal{E}_v^2(\mathbf{m}) - \mathcal{S}(\mathbf{n}) - \mathcal{S}(\mathbf{m})$$

with $\mathbf{m}, \mathbf{n}$ solving the previous saddle-point conditions, at the given value of $\lambda$.

It is also not difficult to see that we must have $\lambda^* = 1/2$, and the exchange happens at a point with $\mathbf{m} = \mathbf{n}$,

$$\mathcal{A}^* = \operatorname*{extr}_{m_+, m_-} \mathcal{E}_h^1(\mathbf{m}) + \mathcal{E}_v^2(\mathbf{m}) - 2\mathcal{S}(\mathbf{m}) \tag{F29}$$

as claimed in the main text.

## Appendix G: Computation of $\tau_{\mathbf{cross}}$

The time to cross from one minima of the free energy to another can be estimated from the tallest energetic barrier that needs to be crossed along the path that connects them [7].

The RBM with weights (E1), generates data around four clusters that can be labeled by the signs of the overlap parameters $m_+, m_-$, as follows: $(m_+, m_-)$, $(m_+, -m_-)$, $(-m_+, m_-)$, and $(-m_+, -m_-)$. Here $m_+, m_-$ are the (positive) roots of the saddle-point equations (7) in the main text.

The values $m_\pm = 0$ are always stationary solution of equations (7). But they do not correspond to minima of the free-energy, instead they are saddle-points. The RBM can jump from one cluster to another, by passing through these saddle-points. We can define the free-energy as a function of the saddle-point overlaps:

$$\mathcal{F} \sim -\frac{1}{N} \ln Z \sim \mathcal{F}_+(m_+, n_+) + \mathcal{F}_-(m_-, n_-) \tag{G1}$$

where

$$-\mathcal{F}_\pm = \alpha_\pm W_\pm m_\pm n_\pm + \frac{1 \pm x}{2} \mathcal{S}_v^\pm(m_\pm) + \alpha_\pm \mathcal{S}_h^\pm(n_\pm) \tag{G2}$$

and

$$\begin{aligned}
\mathcal{S}_v^\pm(m_\pm) &= \min_{\hat{m}_\pm} \left[ -m_\pm \hat{m}_\pm + \ln \cosh(\hat{m}_\pm) \right] + \ln 2 \\
\mathcal{S}_h^\pm(n_\pm) &= \min_{\hat{n}_\pm} \left[ -n_\pm \hat{n}_\pm + \ln \cosh(\hat{n}_\pm) \right] + \ln 2
\end{aligned} \tag{G3}$$

Then, if a saddle-point with non-zero $m_-, n_-$ exists, it is the minimum of the free energy, while the solution $m_- = n_- = 0$, which continues to exist, is a saddle-point. These saddle-points are used to cross from one cluster to another, see Fig. S3. For instance, to cross from $(m_+, m_-)$ to $(m_+, -m_-)$, the least-action path is through $m_- = 0$, and the transition energy cost corresponds to

$$\mathcal{F}_-(m_-) - \mathcal{F}_-(0) \tag{G4}$$

## Appendix H: Parameters used in simulations of crossing rates

Figure 5(b) in the main text simulates the rate at which two stacked RBMs jump between between energetic wells, in two parameter settings:

- $x = 0.8$, $\alpha^1 = 0.4$, $w^1 = 1.2$, $y^1 = 0.2$ for the bottom RBM, and $\alpha^2 = 0.9$, $w^2 = 1.5$, $y^2 = 0.3$ for the top (in black in the Figure).

- $x = 0.8$, $\alpha^1 = 0.4$, $w^1 = 2.0$, $y^1 = 0.4$ for the bottom RBM, and $\alpha^2 = 0.9$, $w^2 = 1.7$, $y^2 = 0.3$ for the top (blue in the Figure).

Then in Figure 5(c) of the main text we considering the time it takes for an RBMs to jump between energetic wells, in two parameter settings:

- $\alpha = 0.7$, $x = 0.2$, $y = 0.1$ and $w = 1.5$ (black).

- $\alpha = 0.7$, $x = 0.4$, $y = 0.2$ and $w = 1.8$ (blue).

$$\mathcal{H}(m) = -\frac{1+m}{2} \ln \left( \frac{1+m}{2} \right) - \frac{1-m}{2} \ln \left( \frac{1-m}{2} \right) \tag{H1}$$

## Appendix I: Supplementary tables

|  | 1st RBM | 2nd RBM | 3rd RBM | 4th RBM |
|---|---|---|---|---|
| MNIST (0/1) | 8 | 4 | 4 | 2 |
| MNIST (full) | 15 | 5 | 4 | 3 |
| 2-D Ising | 10 | 5 | 2 | - |
| Lattice proteins | 30 | 5 | < 1 | - |

TABLE S3. Stacked RBMs training times (in minutes) for each dataset. All RBMs were trained on a NVIDIA RTX A5000 GPU, as described in Section A.

|  | AGS | PT(10) | PT(100) | PT(1000) | ST |
|---|---|---|---|---|---|
| MNIST (0/1) | < 1 | 6 | 10 | 67 | 3 |
| 2-D Ising | 1 | 7 | 14 | 76 | 6 |
| Lattice proteins | 2 | 10 | 17 | 93 | 7 |

TABLE S4. Stacked RBMs sampling times (in minutes) for each dataset and algorithm. The time reported is the time necessary to complete 10000 alternate Gibbs sampling (AGS) steps times 100 MC chains, in each case. Note that different algorithms perform more or less AGS per iteration. A full sweep over all temperatures in PT involves one AGS per temperature, while a full sweep of exchanges in ST involves one AGS per RBM in the stack. Computations performed on a NVIDIA RTX A5000 GPU.

|  | AGS | PT(10) | PT(100) | PT(1000) | ST |
|---|---|---|---|---|---|
| MNIST (0/1) | $6.3 \times 10^3$ | $1.4 \times 10^3$ | $1.1 \times 10^3$ | $1.0 \times 10^3$ | $1.5 \times 10^2$ |
| 2-D Ising | $3.3 \times 10^5$ | $1.5 \times 10^5$ | $5.2 \times 10^4$ | $4.1 \times 10^4$ | $9.7 \times 10^2$ |
| Lattice proteins | $4.0 \times 10^7$ | $2.9 \times 10^7$ | $2.8 \times 10^6$ | $3.1 \times 10^5$ | $5.0 \times 10^2$ |

TABLE S5. Autocorrelation times of different sampling strategies: Alternate Gibbs sampling (AGS), Parallel Tempering (PT, number of intermediate temperatures indicated), and Stacked Tempering (ST). Estimated as the exponential decay of the correlation coefficient in time of the digit classification score (for MNIST0/1), the magnetization (for the Ising model), or the structure probability (for the Lattice proteins). See also Figure S7 for different sizes of the 2-D Ising model (here $L = 32$).

# Appendix J: Additional Figures

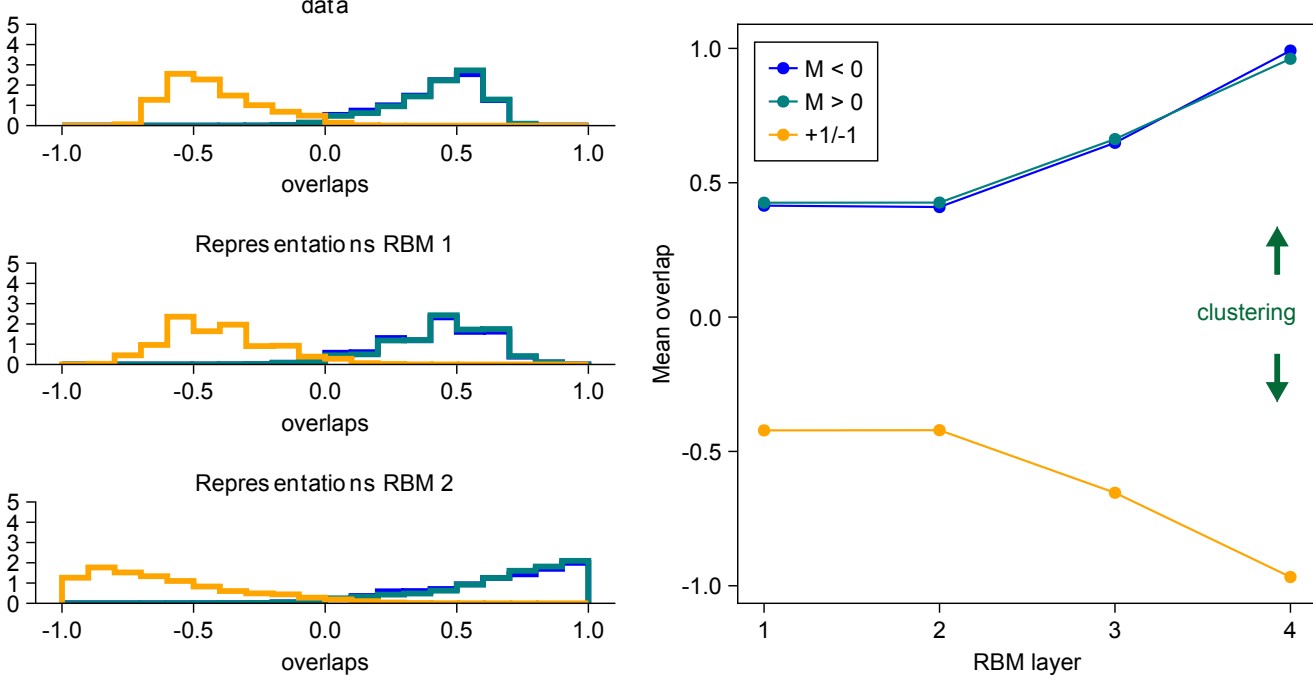

FIG. S4. Overlaps of representations in the 2D Ising model dataset. The left column shows histograms of the Ising spins, the hidden representations of the first RBM, and the hidden representations of the second RBM, in each row. The overlap is defined as the dot product of two configurations, divided by the number of spins. In blue and green we compute overlaps between configurations of the same magnetization sign, while in yellow the overlaps are computed between configurations of opposite magnetization signs. The right column plots the average overlap in each case, for each RBM layer. At deeper levels in the stack, configurations of opposite magnetizations are mapped to representations that become more distant in terms of overlaps, while configurations having magnetizations of the same sign get closer. This geometry is consistent with a clustering of related data points, as indicated by the green arrows.

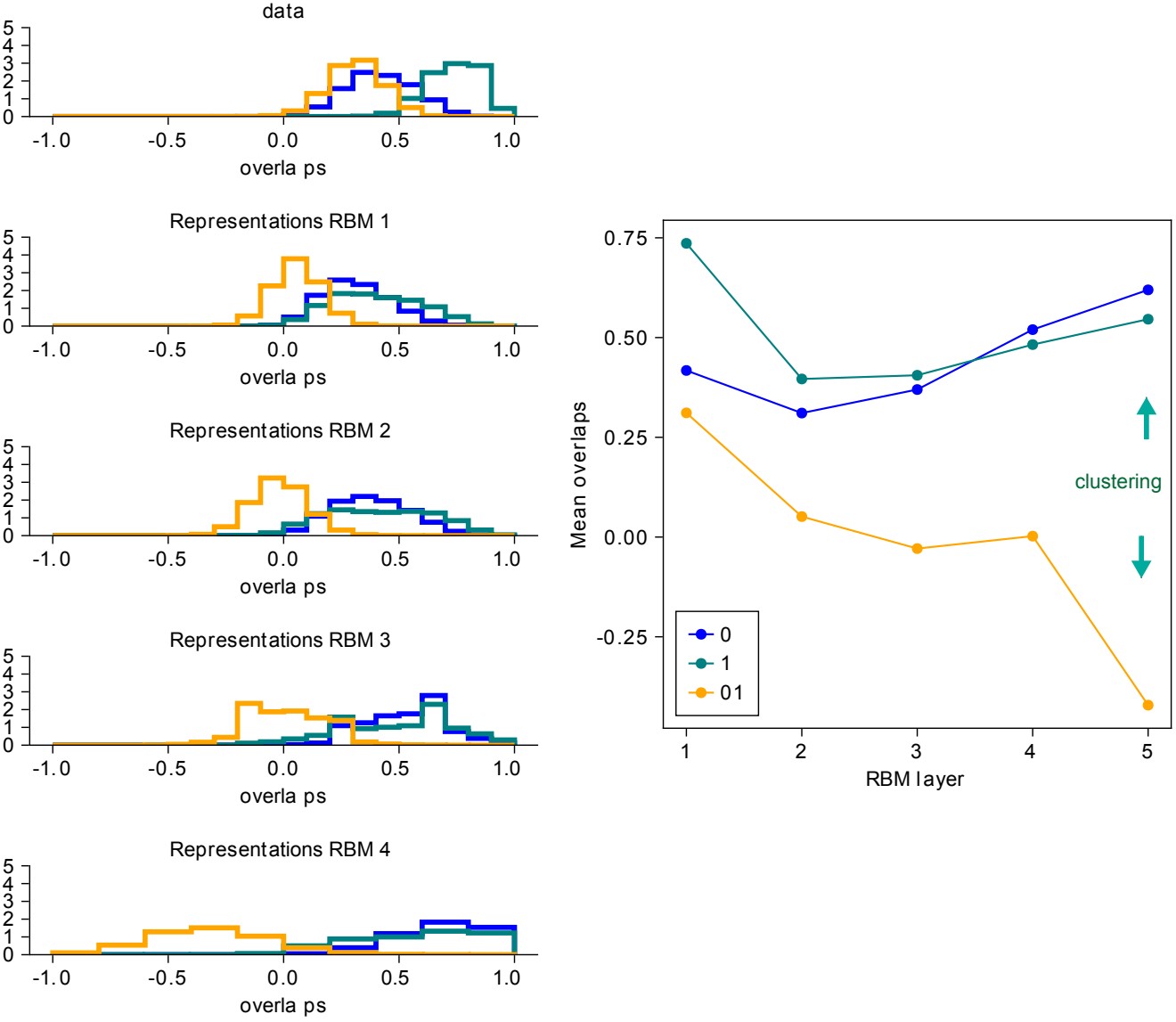

FIG. S5. Overlaps of representations in the MNIST 0/1 dataset. Like Fig. S4, but for the MNIST0/1 dataset. Overlaps of images in the data space are computed by ignoring pixels that are always off across all images. At deeper levels in the stack, images of distinct digits are mapped to representations that become more distant in terms of overlaps, while pairs of images of the same digit are drawn closer to each other. This geometry is consistent with clustering of similar data points, as indicated by the green arrows.

## MNIST (full)

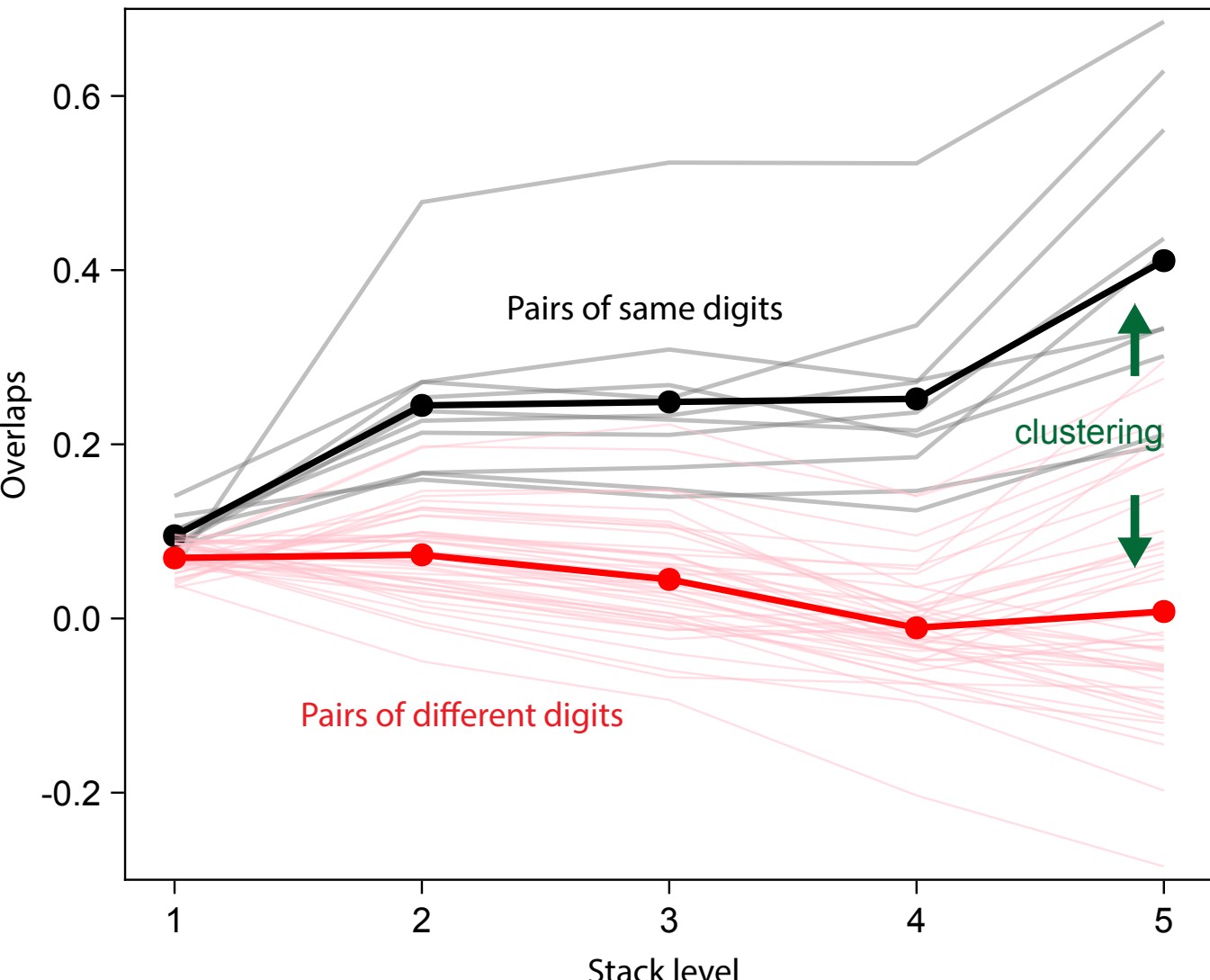

FIG. S6. Overlaps of representations in the MNIST (full) dataset at different levels along the stack of RBMs. Light red lines correspond to pairs of different digits (*e.g.*, overlaps between images of digits 4 and images of digits 5), while gray lines correspond to overlaps between images of the same digit. The thick lines depict the average trend over all pairs of same digits (black) and all pairs of distinct digits (red). At deeper levels in the stack, images of distinct digits are mapped to representations that become more distant in terms of overlaps, while pairs of images of the same digit are drawn closer to each other. This geometry is consistent with clustering of similar data points, as indicated by the green arrows.

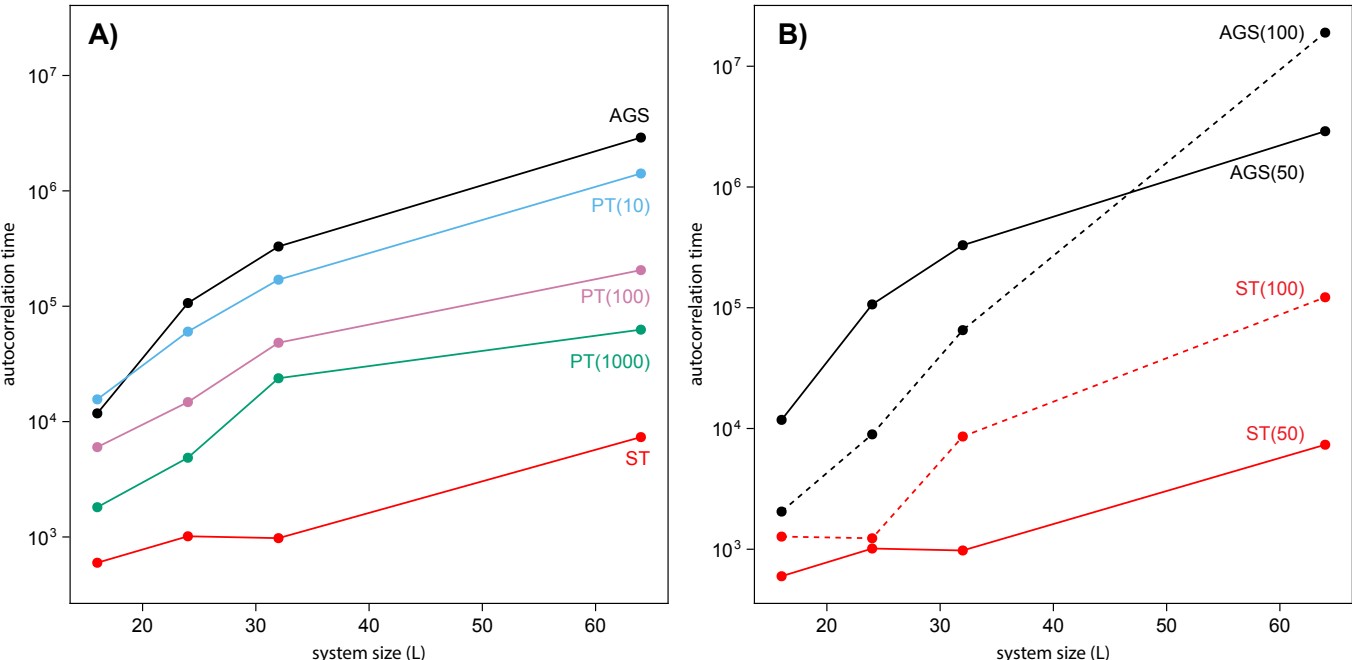

FIG. S7. Autocorrelation times (defined as in Table S5, and in terms of the number of AGS steps) in the 2D-Ising model for different algorithms, as function of the system length, $L$ (the number of spins equals $L^2$ in the square grid). See also Table S5 which gives the results for $L = 32$. Time in the $x$-axis is measured as the number of alternative Gibbs sampling (AGS) steps for each algorithm. For example, in Stacked Tempering (ST) with 4 RBM layers, each iteration of the algorithm involves 4 AGS steps (one per RBM). **A)** Compares ST with parallel tempering (PT) with different numbers of intermediate temperatures, and to AGS. In **A)** and in the simulations shown in main-text, the bottom RBM has $m = 50$ hidden units, which might not be the optimal choice for different system sizes $L$. To illustrate the dependence on $m$, we compare in **B)** the autocorrelations for $m = 50$ and $m = 100$, both for ST and for AGS. Increasing $m$ lets the bottom RBM fit better the Ising model data, but decreases the exchange rates with deeper RBMs (whose hyper-parameters have not changed).

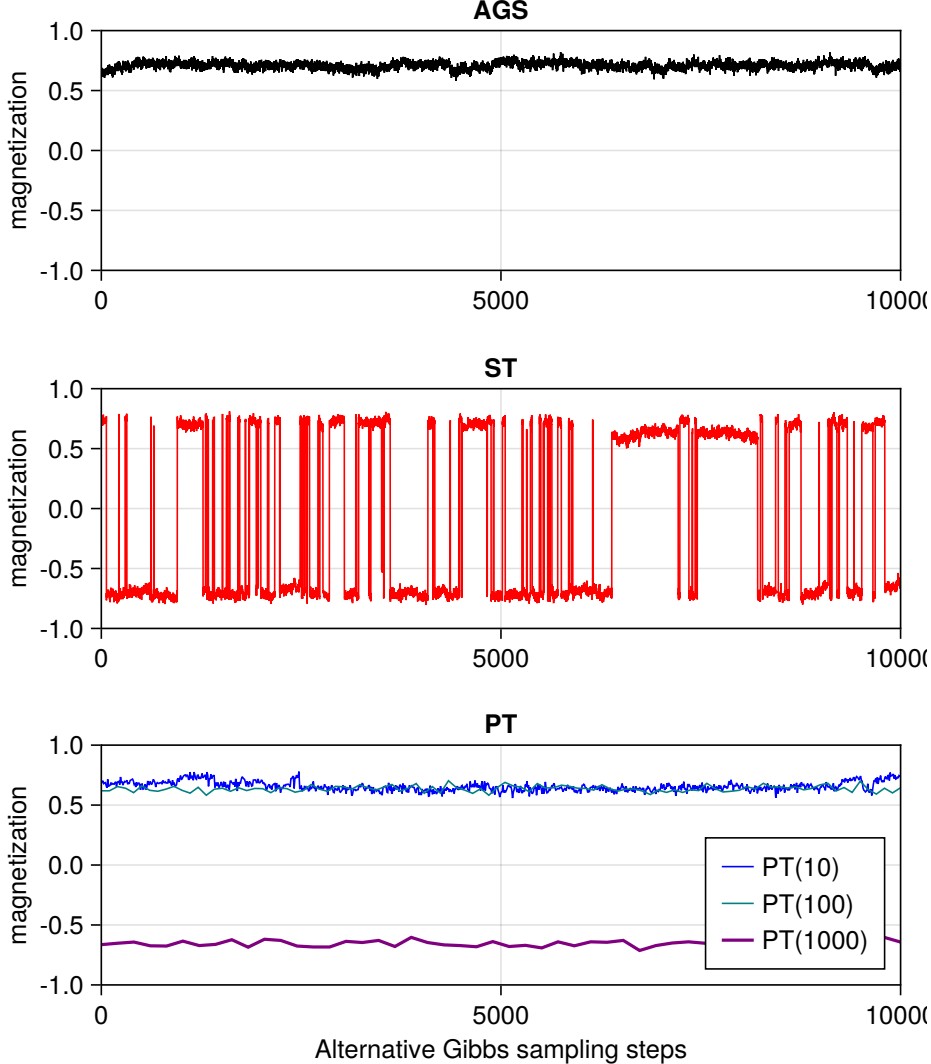

FIG. S8. Example magnetization trajectories as function of time for sampled configurations of the 2D-Ising model ($L = 32$), for different algorithms: AGS (top row), ST (second row), PT (bottom row). Time is measured in terms of the number of Alternating Gibbs sampling (AGS) steps performed in each case. For example, stacked tempering (ST) with 4 RBM layers performs 4 AGS steps per iteration.

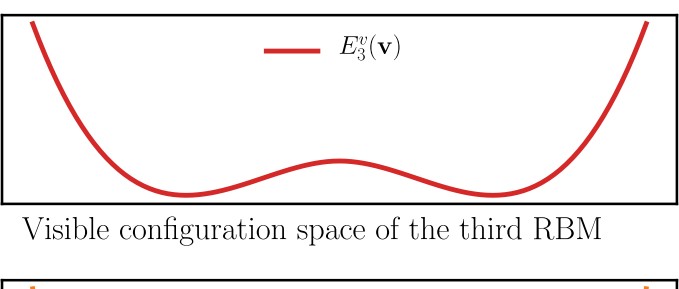

Visible configuration space of the third RBM

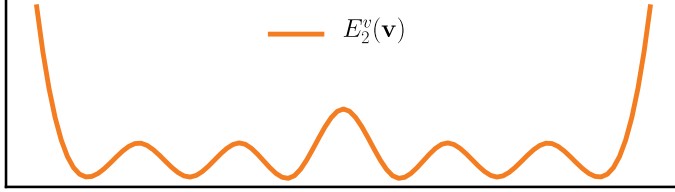

Visible configuration space of the second RBM

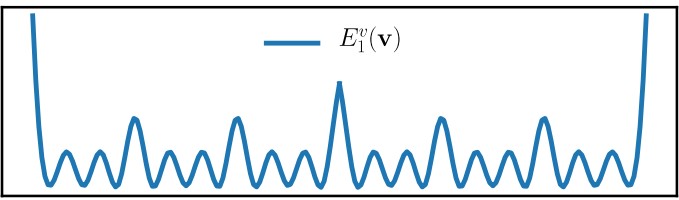

Visible configuration space of the first RBM

FIG. S9. Schematic representation of the landscape $E_1^v(\mathbf{v})$, $E_2^v(\mathbf{v})$ and $E_3^v(\mathbf{v})$ learned by the different RBMs represented in the panel (d). The details of the landscape are progressively smoothing out but in the same time the free energy barriers between the different modes are decreasing.

**Appendix K: Source code**

A Julia package implementing the algorithms discussed in this paper can be found at: `https://github.com/2024stacktemperingrbm/StackedTempering.jl`.

[1] T. Tieleman, in *Proceedings of the 25th international conference on Machine learning* (Association for Computing Machinery, New York, NY, USA, 2008), ICML '08, pp. 1064–1071, ISBN 978-1-60558-205-4, URL https://doi.org/10.1145/1390156.1390290.

[2] D. P. Kingma and J. Ba, arXiv preprint arXiv:1412.6980 (2014).

[3] J. Fernandez-de Cossio-Diaz, S. Cocco, and R. Monasson, Physical Review X **13**, 021003 (2023).

[4] S. Miyazawa and R. L. Jernigain, J Mol Biol **256**, 623 (1996), ISSN 0022-2836, 1089-8638, URL https://europepmc.org/article/med/8604144.

[5] L. Onsager, Phys. Rev. **65**, 117 (1944), URL https://link.aps.org/doi/10.1103/PhysRev.65.117.

[6] P. Mehta and D. J. Schwab, arXiv:1410.3831 [cond-mat, stat] (2014), arXiv: 1410.3831, URL http://arxiv.org/abs/1410.3831.

[7] C. Roussel, S. Cocco, and R. Monasson, Phys. Rev. E **104**, 034109 (2021), URL https://link.aps.org/doi/10.1103/PhysRevE.104.034109.