# OpenReview forum: "Accelerated Sampling with Stacked Restricted Boltzmann Machines"
_ICLR.cc/2024/Conference — ICLR 2024 poster_

### Official Review · Reviewer_mREg · 2023-10-31

**Soundness:** 3 good
**Presentation:** 4 excellent
**Contribution:** 3 good
**Rating:** 6
**Confidence:** 4

**Summary:**

The manuscript introduces a stacked RBM parallel tempering training, in which sampling is improved by swapping configurations of hidden and visible units in the vertical stack.

**Strengths:**

The paper introduces an interesting idea, that allows to sample more efficiently from RBMs, without the need to specify effective temperatures, as in Parallel Tempering

**Weaknesses:**

The numerical experiments do not allow to understand the improvement over more standard Parallel Tempering

**Questions:**

Can the authors show the autocorrelation times of some observable (for example, on the Ising model, the magnetization at the critical point) ? It is hard to understand, from the proposed plots, whether the scheme introduced is faster than standard PT

The simplified theoretical analysis of mixing and swapping times is very interesting, but as far as I can see there is no direct comparison to simulation data. It would be desirable to have such a comparison, in order to understand whether the qualitative aspects of this analysis carry on to the more complex algorithm

---

> ### Author Response · Authors · 2023-11-15
>
> We thank the reviewer for the comments.
>
> > The numerical experiments do not allow to understand the improvement over more standard Parallel Tempering.
>
> Fig. 3 shows numerical simulations in three different datasets (MNIST, Lattice Proteins, 2D Ising model), where stacked tempering (ST) outperforms parallel tempering (PT) in terms of the rate of transitions effected between minima of the energy landscape. To compare different algorithms in a fair way, we count the number of Alternate Gibbs sampling (AGS) steps. Note that different algorithms perform different numbers of AGS per iteration. For example, PT does one AGS per temperature, and a full sweep over all temperatures involves many AGS. In Fig. 3, the x-axis reports the number of AGS steps (we have modified the caption to emphasize this point). The figure demonstrates that ST transitions more frequently between modes than vanilla AGS or PT.
>
> > Can the authors show the autocorrelation times of some observable (for example, on the Ising model, the magnetization at the critical point)? It is hard to understand, from the proposed plots, whether the scheme introduced is faster than standard PT
>
> We have computed the autocorrelation times for the Ising model, MNIST0/1, and for the Lattice proteins, in Supplementary Table S4. Generally, the autocorrelation is between one or two orders of magnitude smaller for ST than for PT or AGS, as expected from Fig. 3.
>
> > The simplified theoretical analysis of mixing and swapping times is very interesting, but as far as I can see there is no direct comparison to simulation data. It would be desirable to have such a comparison, in order to understand whether the qualitative aspects of this analysis carry on to the more complex algorithm
>
> The simplified analysis is meant to demonstrate the existence of a regime where stacked tempering has an exponential advantage over local sampling techniques such as Metropolis or AGS. Moreover, the analysis provides a mechanistic explanation of this advantage, related to an interplay of clustering and separation of the representations extracted by the model. Figure 5 and SI2, 3 show numerical simulations of the simplified model, in excellent agreement with the theoretical calculations.
>
> We have added SI figures S4, S5, S6, which illustrate how representations along the stack of RBMs cluster and separate, for MNIST0/1, MNIST full, and Ising datasets. Representations at deeper levels of the stack bring closer together data points that are similar, while driving apart more dissimilar points. Our simplified theoretical analysis then allows us to hypothesize that these changes in the representation geometry facilitate sampling.

---

> > ### Comment · Reviewer_mREg · 2023-11-16
> > **Performance comparison to PT**
> >
> > > We have computed the autocorrelation times for the Ising model, MNIST0/1, and for the Lattice proteins, in Supplementary Table S4. Generally, the autocorrelation is between one or two orders of magnitude smaller for ST than for PT or AGS, as expected from Fig. 3.
> >
> > Thank you, apologies for the oversight if the Table was already there, that is what I was looking for. I think that the analysis on the autocorrelation times should take precedence over the more qualitative analysis of Fig. 3, but that is just my own taste.
> >
> > Generally speaking, it is nice to see a significant reduction in the autocorrelation times when compared to PT with a reasonable number of copies. However, to be entirely rigorous here, one would need to have at least an idea of the scaling of correlation times (at the critical temperature) with system size. If the authors could add that, it would nicely complete the assessment over PT also in terms of scaling.
> >
> > Also, while I understand the authors emphasize applications based on unsupervised learning, it would be nice to maybe add a comment on the fact that their approach is quite general and can can be used to improve sampling of an RBM more broadly. I.e., those applications in which there is no external data, but weights of an RBM are given, and the task is sampling from those. These are more common for example in applications to quantum physics.

---

> > > ### Author Response · Authors · 2023-11-21
> > >
> > > We thank the reviewer for their comments to our first round of revision.
> > >
> > > > Generally speaking, it is nice to see a significant reduction in the autocorrelation times when compared to PT with a reasonable number of copies. However, to be entirely rigorous here, one would need to have at least an idea of the scaling of correlation times (at the critical temperature) with system size. If the authors could add that, it would nicely complete the assessment over PT also in terms of scaling.
> > >
> > > We have computed autocorrelation times for the 2D Ising system at β=0.44 (approximately the critical temperature), for different sizes (L = 16, 24, 32 and 64). The results are shown in SI Fig. S7. We find, consistently with the results already reported for L = 32, that ST is faster than PT and AGS.
> > >
> > > However, we believe more extensive numerical tests are needed to properly assess the scaling of ST with system size at criticality. The reason is that, in the numerical experiments reported above, we have not tuned the hyper-parameters of the RBM stack (e.g., number of RBM levels, numbers of hidden units in each RBM, regularizations, etc.) for each value of L. Fig. S7(A) shows only one curve for ST with fixed hyper-parameters (in particular, m=50 hidden units for the bottom RBM). We expect that the optimal number of hidden units in the stack should exhibit a non-trivial scaling with L, related to the dependence of the correlation length upon size at the critical temperature in the 2D Ising model. To illustrate the importance of the choice of m, Fig. S7(B) shows the autocorrelation times when ‘m’ is changed from 50 to 100, both for ST and AGS. Increasing ‘m’ lets the bottom RBM fit better the Ising model data, but decreases the exchange rates with deeper RBMs (whose hyper-parameters have not changed). We plan to address the important question of the optimal choice of hyper-parameters for the stack in future work; this point is now made explicit in the Discussion.
> > >
> > > > Also, while I understand the authors emphasize applications based on unsupervised learning, it would be nice to maybe add a comment on the fact that their approach is quite general and can can be used to improve sampling of an RBM more broadly. I.e., those applications in which there is no external data, but weights of an RBM are given, and the task is sampling from those. These are more common for example in applications to quantum physics.
> > >
> > > We agree RBMs have many applications besides unsupervised learning. We have added references to make this point in the Discussion (as space permits).

---

### Official Review · Reviewer_fKHh · 2023-11-04

**Soundness:** 3 good
**Presentation:** 3 good
**Contribution:** 2 fair
**Rating:** 6
**Confidence:** 4

**Summary:**

This paper introduces a method called Stacked Tempering (ST) that applies the ideas of deep tempering to restricted Boltzmann machines (RBM). The authors propose to learn stacks of nested RBMs, where the representations of one RBM are used as "data" for the next one in the stack. By exchanging configurations between RBMs, the ST method allows for fast transitions between different modes of the data distribution. The paper provides analytical calculations of mixing times and demonstrates the efficiency of the ST method compared to standard Monte Carlo methods on several datasets.

**Strengths:**

- This paper introduces a new approach called Stacked Tempering (ST) for sampling from Restricted Boltzmann Machines (RBMs).
  - ST learns nested RBM stacks by using the representation of one RBM as the "data" for the next RBM.
- Efficiency of the ST method is demonstrated through experiments on multiple datasets including MNIST, in-silico Lattice Proteins, and 2D-Ising model.
- This paper provides the first theoretical analysis supporting the use of deep representations for improving mixing in RBMs, inspired by previous research on deep tempering.
  - Obtained analytical results are interesting.

**Weaknesses:**

- Learning nested RBMs seems costly compared to simple parallel tempering approach where additional models are not required.
- This method is only applicable to RBMs.

**Questions:**

- Is it possible to extend this method for sampling from other energy-based models (e.g., deep Boltzmann machines)?

---

> ### Author Response · Authors · 2023-11-15
>
> We thank the reviewer for the comments.
>
> > Learning nested RBMs seems costly compared to simple parallel tempering approach where additional models are not required.
>
> Although parallel tempering (PT) requires no additional training, it requires several (usually hundreds or thousands) copies of the RBM at different temperatures that exchange configurations. The temperatures are also not easy to set a priori. Stacked tempering (ST) requires additional training of a few models (our experiments were performed with 4 RBMs at most). In addition, training the deeper models is faster because: 1) the landscapes are smoother at deeper levels of the stack, and 2) the layers of the stack shrink and therefore these models are trained on data of smaller dimensionality. Table S3 in the Appendix now reports the training times. As can be seen from table S4, the cumulative training of the deeper RBMs takes a time comparable to the training time of the bottom RBM.
>
> Finally, we note that the total sampling time of stacked tempering can be decomposed as training time, which is a constant offset, plus sampling time, which is proportional to the number of generated samples. The stack needs to be trained once only.
>
> > This method is only applicable to RBMs.
>
> Although our formulation and numerical examples use RBMs, the method is more general. See answer below for examples.
>
> > Is it possible to extend this method for sampling from other energy-based models (e.g., deep Boltzmann machines)?
>
> We believe several extensions are possible, but numerical experiments are out of scope for the present work. We speculate briefly on possible extensions.
>
> * A deep Boltzmann machine (DBM) is a particular case of an RBM (since it is bipartite, with even and odd layers on each partition), and therefore the method we have described applies in principle without modifications. It would be interesting to carry out numerical experiments to assess whether stacked tempering (ST) can help training DBMs, which are known to be computationally difficult in practice.
> * We have presented a simple formulation where the stack is trained layer by layer. Training can also be carried out “in parallel” for all RBMs in the stack, allowing gradients to flow from bottom layers to the top using a reverse KL divergence. This would allow exchanged samples to be used during training of the bottom layers.
> * More generally, energy-based models (such as RBM) map data configurations x to an energy E(x), eventually defining a probability over the data space P(x) = exp(-E(x))/Z. As in Refs. [1,2] E(x) can be parameterized by some deep neural network (NN), and P(x) can be sampled by local Langevin dynamics which can get stuck in rugged landscapes. Focusing on the activity h of an intermediate representation layer, we can write E(x) = F(h) = F(H(x)) as a composition where h = H(x) denotes the mapping from the data to the intermediate representation layer. We can train another generative NN for the inverse map x=g(h), by training an energy-based model with the energy E(g(h)) to fit the data h = H(x), with respect to g. The configurations h from the two NNs can then be exchanged according to Metropolis rule for the energy cost E(g(h1)) + F(h2) - E(g(h2)) - F(h1). In this manner the intermediate representations h are exploited to propose global moves in the data space. The stacked tempering method we have described can be regarded as the simplest implementation of this idea, where the mappings g(h), E(x), F(h), are naturally defined for the RBMs.
>
> References:
> * Song & Kingma (2021) [arXiv:2101.03288](https://arxiv.org/abs/2101.03288)
> * Du, Mordatch (2019) [NeurIPS](https://papers.nips.cc/paper_files/paper/2019/hash/378a063b8fdb1db941e34f4bde584c7d-Abstract.html)

---

> > ### Comment · Reviewer_fKHh · 2023-11-24
> > **Response to the authors' rebuttal**
> >
> > Thank you for replying to my review.
> >
> > My concern is mainly addressed through the rebuttal, and I feel more positive about this paper than before the rebuttal.
> > Although I keep my score to 6, I raise my confidence to 4.

---

### Official Review · Reviewer_4ET5 · 2023-11-20

**Soundness:** 4 excellent
**Presentation:** 3 good
**Contribution:** 3 good
**Rating:** 8
**Confidence:** 3

**Summary:**

The paper proposes a new sampling scheme for RBMs that utilizes a hierarchy of RBMs of decreasing latent and visible dimensions to encourage learning a smoother version of the data distribution that captures more global aspects of the distribution in a manner akin to temperature annealing in parallel tempering. Their training approach involves sampling the latent variables at a given layer using alternating (between the latent and visible variables) Gibbs sampling (AGS) and then using the latent variable samples as training data for the next layer up in the hierarchy. Sampling involves using AGS to generate latent variables in a layer and then swapping the latent variables of one layer with the visible variables of the next one up using an acceptance probability. Their experiments on MNIST, a protein folding problem and the 2-D Ising model demonstrate how their approach can sample modes much quicker than AGS and parallel tempering. They also give a theoretical analysis in the overparameterized regime showing how different settings of the hyperparameters (regularization weight and ratio of hidden-to-visible variables) induces different representational regimes of the model. They also perform an analysis of the mixing time of their scheme indicating how it matches empirical data

**Strengths:**

- The main strengths of the paper are the results which indicate their approach's superior sampling performance compared to other reasonable baselines of parallel tempering and AGS.
- The explanation of their approach is clear to understand and is given succinctly.
- The theoretical analysis gives good insight and interpretation of the results of their approach as well as the consequences of different settings of the hyperparameters. In addition, the fact that the relations established for the mixing times (as in Figure 5) match the empirical results adds confidence in the correctness of their analysis

**Weaknesses:**

- As their approach is intended to speed up sampling of RBMs a figure/table demonstrating how the real world sampling time is approved compared to the parallel tempering and AGS baselines would give the reader a better sense of how this method fares practically.
- Adding figures similar to Fig 3(d-h) for the Ising model results in the main text would give the reader a better understanding of the results on this problem
- A conceptual comparison to deep Boltzmann machines would be a great addition as it would make clear how their approach differs from sampling schemes for deep Boltzmann machines.

**Questions:**

- No questions that need clarifying

---

> ### Author Response · Authors · 2023-11-21
>
> We thank the reviewer for their comments.
>
> > As their approach is intended to speed up sampling of RBMs a figure/table demonstrating how the real world sampling time is approved compared to the parallel tempering and AGS baselines would give the reader a better sense of how this method fares practically.
>
> We have added table S4 in Supplementary materials during revision, with sampling times (in minutes) in our experiments for different algorithms and datasets. Overall we find that ST is faster than AGS or PT.
>
> > Adding figures similar to Fig 3(d-h) for the Ising model results in the main text would give the reader a better understanding of the results on this problem
>
> We have added Figure S8 in supplementary showing example trajectories of the magnetization in the Ising model during sampling with the different algorithms. ST samples more easily configurations of alternate magnetization than AGS or PT.
>
> > A conceptual comparison to deep Boltzmann machines would be a great addition as it would make clear how their approach differs from sampling schemes for deep Boltzmann machines.
>
> We agree that applications to deep Boltzmann machines (DBM) would be interesting and we would like to explore this in future work. A DBM can be regarded as a particular case of an RBM, in  which “visible” and “hidden” layers are made of the even and odd layers of the DBM. Therefore, stacked tempering (ST) should apply without modification. Whether ST can help training DBMs is an interesting open question, which we now briefly mention in Discussion.

---

> ### Comment · Reviewer_4ET5 · 2023-11-22
>
> Thank you for addressing my questions/weaknesses. I'll keep my score the same

---

### Author Response · Authors · 2023-11-15

We thank the reviewers for their comments on our work. We have addressed their questions and weakness comments below.

---

### Meta-Review · Area_Chair_6Fuz · 2023-12-12

**Metareview:**

Summary: The article presents a sampling algorithm for RBMs building on parallel and deep tempering.

Strengths: Three referees are generally positive about the submission. Referees appreciated the positive comparison against baselines, clear explanation of the approach and theoretical analysis.

Weaknesses: Initial reviews asked for data on sampling times and certain experiments, some of which were added during the rebuttal period. Further concerns were the possible cost and limitations of the proposed method. The author responses could address some of these concerns giving the referees increased confidence on their favorable assessment. Referees found that comparison or application to deep Boltzmann machines could have helped clarify the differences of the proposed method to existing methods.

In view of the positive reception and favorable outcome of the discussion, I am recommending accept. The authors are encouraged to work on adding more extensive numerical tests on items discussed during the discussion period.

**Justification For Why Not Higher Score:**

The proposed methods have a focus on RBMs and exploration of applications to broader families have yet to be carried out, a natural item that could have further strengthened the work.

**Justification For Why Not Lower Score:**

The article has generally positive responses from the reviewers. Initial concerns were adequately addressed during the discussion period.

---

### Decision · Program_Chairs · 2024-01-16

Accept (poster)